# Improved clearing method contributes to deep imaging of plant organs

Yuki Sakamoto[1,2], Anna Ishimoto[3], Yuuki Sakai[4], Moeko Sato[5], Ryuichi Nishihama [3,6], Konami Abe[3], Yoshitake Sano [1,3], Teiichi Furuichi [1,3], Hiroyuki Tsuji [5], Takayuki Kohchi [6] & Sachihiro Matsunaga [1,3,7✉]

Tissue clearing methods are increasingly essential for the microscopic observation of internal tissues of thick biological organs. We previously developed TOMEI, a clearing method for plant tissues; however, it could not entirely remove chlorophylls nor reduce the fluorescent signal of fluorescent proteins. Here, we developed an improved TOMEI method (iTOMEI) to overcome these limitations. First, a caprylyl sulfobetaine was determined to efficiently remove chlorophylls from *Arabidopsis thaliana* seedlings without GFP quenching. Next, a weak alkaline solution restored GFP fluorescence, which was mainly lost during fixation, and an iohexol solution with a high refractive index increased sample transparency. These procedures were integrated to form iTOMEI. iTOMEI enables the detection of much brighter fluorescence than previous methods in tissues of *A. thaliana*, *Oryza sativa*, and *Marchantia polymorpha*. Moreover, a mouse brain was also efficiently cleared by the iTOMEI-Brain method within 48 h, and strong fluorescent signals were detected in the cleared brain.

[1] Imaging Frontier Center, Organization for Research Advancement, Tokyo University of Science, 2641 Yamazaki, Noda, Chiba 278-8510, Japan. [2] Department of Biological Sciences, Graduate School of Science, Osaka University, Machikaneyama-cho 1-1, Toyonaka, Osaka 560-0043, Japan. [3] Department of Applied Biological Science, Faculty of Science and Technology, Tokyo University of Science, 2641 Yamazaki, Noda, Chiba 278-8510, Japan. [4] Department of Biology, Graduate School of Science, Kobe University, Kobe 657-8501, Japan. [5] Kihara Institute for Biological Research, Yokohama City University, Maioka 641-12, Totsuka, Yokohama 244-0813, Japan. [6] Graduate School of Biostudies, Kyoto University, Kyoto 606-8502, Japan. [7] Department of Integrated Biosciences, Graduate School of Frontier Sciences, The University of Tokyo, 5-1-5 Kashiwanoha, Kashiwa, Chiba 277-8562, Japan. ✉email: sachi@edu.k.u-tokyo.ac.jp

The development of microscopes, dyes, fluorescent proteins (FPs), and sample preparation methods have enabled observation of bright and high-resolution microscopic images, which is the driving force to unravel the mystery of life. In particular, FPs are indispensable tools to visualize cellular components, including organelles, proteins, nucleic acids, and small molecules, and provide information such as temperature, pH, and voltage[1–3]. However, it is often difficult to sufficiently detect the fluorescent signals of FPs emitted from the internal regions of three-dimensionally thick tissues because autofluorescent pigments absorb the light and some cell components, which refract or reflect light, scatter and disturb the signals. Several clearing methods to elute pigments from cells and adjust the refractive index (RI) through the specimen to the mounting medium have been developed to observe the tissue deeply embedded in an organ.

The clearing methods developed for animal tissues, including ScaleS, CUBIC, PACT, and SeeDB2, can maintain the fluorescence of FPs in highly transparent tissues[4–8]. ScaleS uses the hyperhydration effect of urea to make animal tissues transparent, CUBIC and PACT employ effective detergents to remove obstacles, and SeeDB2 uses iohexol solution for transparency. For plant tissues, Scale-based method[9], ClearSee[10,11], PEA-CLARITY[12], and TOMEI[13–15] were developed as optical clearing methods and preserve the fluorescence of FPs. In the Scale-based method, *Nicotiana benthamiana* leaves, transiently expressing the fluorescent actin marker mTalin-Citrine, were fixed with 4% paraformaldehyde and then incubated in a solution containing 6 M urea, 30% (v/v) glycerol, and 0.1% (v/v) Triton X-100 for 1 to 3 weeks[9]. After clearing, chlorophylls were eluted entirely from leaves, and both fine and thick actin bundles were observed in leaf epidermal and mesophyll cells. In the ClearSee, seedlings, leaves, roots, pistils, and stems of *A. thaliana* and gametophytes of *Physcomitrium patens* were fixed with 4% paraformaldehyde and then treated with ClearSee solution [10% (w/v) xylitol, 15% (w/v) sodium deoxycholate, and 25% urea] for 4 days to 4 weeks[10]. After clearing, chlorophylls were wholly removed, and seven FPs (mTFP1, sGFP, mClover, Venus, mCitrine, tdTomato, and mApple) were visualized in *A. thaliana*. In the PEA-CLARITY, leaves of *A. thaliana* and *N. benthamiana* were fixed in a hydrogel solution containing 4% paraformaldehyde[12]. Hydrogel, including the sample, was polymerized and then treated with a 4% SDS solution for 4 to 6 weeks. After clearing, chlorophylls were removed entirely, and the fluorescence of GFP and CFP was detected. We have reported two types of TOMEIs; TOMEI-I and TOMEI-II[13]. When observing FPs, TOMEI-II should be used because TOMEI-I results in FP quenching. For the TOMEI-II method, leaves and root knots of *A. thaliana* were fixed with 4% paraformaldehyde and then treated with increasing concentrations of 2,2′-thiodiethanol (TDE) in steps to avoid drastic osmotic changes. The final concentration of TDE may be as high as 97% (v/v). At this concentration, the RI of the TDE solution is 1.52, which is optimal for observation with an oil-immersion lens. The final TDE concentration should be considered carefully because a solution of less than 80% (v/v) TDE has a limited quenching effect on FPs[16–18], whereas a solution of greater than 80% (v/v) TDE largely quenches FPs. Although the TOMEI-II method requires only 3 h to complete, it did not wholly remove chlorophylls from the leaves.

All four methods substantially ameliorated the optical properties of the specimens; however, each method has drawbacks. The Scale-based method, ClearSee, and PEA-CLARITY totally remove chlorophylls from the plant tissues, but they are time-consuming. Although TOMEI-II rapidly clears plant tissues and organs[14,15], certain autofluorescent pigments remain in plant tissues after clearing, and a high TDE concentration reduces the fluorescence intensities of the FPs[13,19].

To overcome the problems of TOMEI-II, we modified all steps of the TOMEI method, including decolorization, FP reactivation, and mounting, and propose here an improved version of TOMEI, which we designate iTOMEI. We successfully performed deep imaging in plant tissues of *A. thaliana*, *Oryza sativa*, and *Marchantia polymorpha* as well as mouse brain tissue using iTOMEI.

## Results

**Caprylyl sulfobetaine elutes chlorophylls without GFP quenching.** In a previous study, we developed the plant tissue-clearing method TOMEI-II. The TOMEI-II protocol is as follows: Step 1: A sample is fixed with 4% paraformaldehyde in PBS buffer (pH 7.0) for 1 h. Step 2: The fixed sample is treated sequentially with 10, 30, 50, and 70% (v/v) TDE solution supplemented with 0.0025% (w/v) propyl gallate for 10 min each. Step 3: The sample is incubated in a final concentration of TDE solution supplemented with 0.0025% (w/v) propyl gallate for 1 h. The final concentration of the TDE solution depends on FPs, but the highest concentration is 97% (v/v) TDE (RI = 1.52). Although the autofluorescence of chlorophylls derived from chloroplasts is an obstacle for the deep imaging of plant tissues, TOMEI-II does not completely remove chlorophylls.

To identify a reagent for the efficient removal of chlorophylls, we screened 19 detergents. For each detergent, 2-week-old plants of *A. thaliana* were fixed with 2% formaldehyde (FA) in PBS buffer for 1 h and then incubated in solutions supplemented with 10% (w/v) detergent for 24 h (Supplementary Table 1). The absorbances of chlorophylls, which were eluted from the seedlings to the detergent solutions, were measured using an absorption spectrophotometer. We determined that caprylyl sulfobetaine (#7) and sodium deoxycholate (#3) were the best and second-best detergents, respectively, for the efficient elution of chlorophylls (Fig. 1a). To investigate their effects on GFP fluorescence, wild-type plants and plants overexpressing GFP were fixed with 2% FA in PBS buffer for 1 h and then incubated with a 10% (w/v) caprylyl sulfobetaine or 10% (w/v) sodium deoxycholate solution for 24 h. Fluorescence images of the same plant were captured before and after the treatment using a fluorescence stereomicroscope. A hypocotyl was selected as the ROI for the measurement of GFP fluorescence intensity. Both detergents enhanced GFP fluorescence after incubation (Fig. 1b). This enhancement was consistent with a previous report for ClearSee[10]. The presence of chlorophylls may decrease the detectable GFP fluorescence in the hypocotyl. Our assessment determined that caprylyl sulfobetaine is a prime candidate for chlorophyll elution without reducing GFP fluorescence.

To enhance the effects of caprylyl sulfobetaine on chlorophyll clearance, we tested the combinations of caprylyl sulfobetaine with Triton X-100, which is used the Scale-based method (Fig. 1c), sodium deoxycholate, which is used in ClearSee (Fig. 1d), and urea, which is used in the Scale-based method and ClearSee (Fig. 1e). We prepared solutions of 10% (w/v) caprylyl sulfobetaine, Triton X-100, and sodium deoxycholate, as well as mixtures of 5% (w/v) caprylyl sulfobetaine plus 5% (w/v) Triton X-100, 5% (w/v) caprylyl sulfobetaine plus 5% (w/v) sodium deoxycholate, and 10% (w/v) caprylyl sulfobetaine plus 25% (w/v) urea. Two-week-old plants were fixed with 2% FA in PBS buffer for 1 h and then incubated in the above detergent solutions independently for 24 h. The mixtures failed to enhance chlorophyll elution compared with the 10% (w/v) caprylyl sulfobetaine solution, suggesting that incubation with caprylyl sulfobetaine alone is most efficient. Subsequently, the chlorophyll elution efficiency and GFP fluorescence were measured after incubation with PBS buffer supplemented with 10, 20, or 30% (w/v) caprylyl sulfobetaine. The 20 and 30% (w/v) solutions showed higher chlorophyll elution efficiency levels than the 10% (w/v) solution (Fig. 1f). In comparison, the

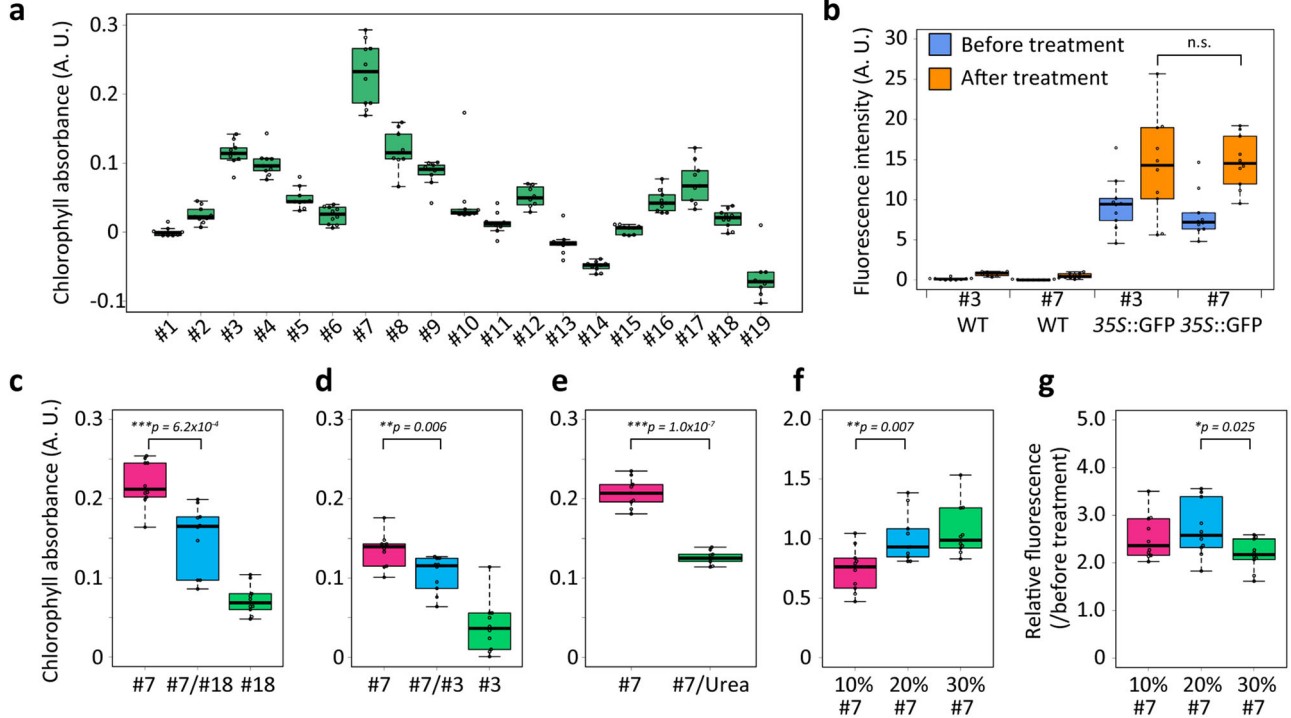

**Fig. 1 Consideration of decolorization conditions. a** Two-week-old plants were fixed with 2% FA and then incubated in 10% (w/v) #1 to #19 detergent solutions (described in Supplementary Table 1) for 24 h. Chlorophyll absorbance in the detergent solutions was measured using an absorption spectrophotometer. The sample size was indicated in Supplementary Data 1. **b** Fluorescence intensity quantified from fluorescence images of the wild type and plants expressing GFP before and after incubation in 10% (w/v) #3 and 10% (w/v) #7 solution. **c–e** Two-week-old plants were fixed with 2% FA and then incubated in the solutions of 10% (w/v) #3, #7, and #18 and in the mixtures of 5% (w/v) #7 plus 5% (w/v) #18, 5% (w/v) #7 plus 5% (w/v) #3, and 10% (w/v) #7 plus 25% (w/v) urea for 24 h and then chlorophyll absorbance in the detergent solutions was measured. **f** Twenty-day-old plants were fixed with 2% FA and then incubated in 10, 20, and 30% (w/v) #7 solution for 24 h. The chlorophyll absorbance in the detergent solutions was measured. **g** Relative fluorescence intensity is calculated as fluorescence intensity in the plant after treatment over that before treatment. The fluorescence intensity was measured from fluorescence images of plants expressing GFP before and after incubation in 10, 20, and 30% (w/v) #7 solution for 24 h. Statistical significance was evaluated using the two-sided Welch's $t$-test. $n = 9$ plants (**e**) and 10 plants (**b**, **c**, **d**, **f**, **g**).

30% (w/v) solution exhibited a lower fluorescence intensity than the 20% (w/v) solution (Fig. 1g), indicating that 20% (w/v) caprylyl sulfobetaine was optimal for our clearing technology.

**Alkaline buffers chemically reactivate GFP fluorescence in plant tissues.** Chemical reactivation (CR) by alkaline treatments enables the recovery of reduced FP fluorescence in resin-embedded in animal tissue specimens[20,21]. We tested the effects of CR on GFP fluorescence in *A. thaliana*. Seedlings of wild-type plants and GFP-expressing plants were fixed with 2% FA in PBS buffer for 1 h and then treated independently with 0.5 mM tri-sodium phosphate buffer at pH 10.7 and 100 mM sodium carbonate buffer at pH 11.3 for 2 h. Both treatments significantly enhanced GFP fluorescence (Fig. 2a), demonstrating that CR recovered the reduced fluorescence of GFP in plants. In addition, reactivated GFP fluorescence after CR was maintained in the PBS buffer for 3 h (Fig. 2a). To determine the optimal buffer pH for CR, we prepared buffers ranging in pH from 8.0 to 12.0. Sodium dihydrogen phosphate and disodium hydrogen phosphate were mixed at 5:95 and 0:100 ratios to produce pH 8.0 and pH 9.0 buffers, respectively. Trisodium phosphate was diluted with water to produce pH 10.0 to pH 12.0 buffers. The pH levels of all the buffers were measured using a pH meter. Seedlings were fixed with 2% FA in PBS buffer for 1 h and then incubated in the pH 7.4 to pH 12.0 buffers for 2 h. Buffers at pH 8.0 to 11.0 recovered the same degree of GFP fluorescence (Fig. 2b); therefore, we chose the pH 8.0 buffer for our clearing method.

Next, to simplify the process and reduce the procedure time, we attempted to combine decolorization with CR. Seedlings expressing GFP were fixed with 2% FA in PBS buffer for 1 h and then incubated in PBS buffer at pH 7.4, or 100 mM sodium phosphate buffer at pH 8.0, supplemented with 20% (w/v) caprylyl sulfobetaine for 24 h. In addition, the fixed seedlings were incubated in PBS buffer at pH 7.4 with 20% (w/v) caprylyl sulfobetaine for 24 h and sequentially incubated in 100 mM sodium phosphate buffer at pH 8.0 for 2 h (Fig. 2c). Incubation in the pH 8.0 buffer, compared within the pH 7.4 buffer, resulted in a higher GFP fluorescence intensity that was comparable to the serial treatment at pH 7.4 for 24 h and pH 8.0 for 2 h (Fig. 2c). Thus, decolorization and CR were performed simultaneously in this method.

**Iohexol enables the clearing of plant organs without FP quenching.** We searched for a mounting reagent superior to TDE, which is used for TOMEI-II. Iohexol dramatically improves the transparency of mouse brain tissue without FP quenching[7]. We investigated whether iohexol improved the clarity without affecting GFP fluorescence in plant tissues. To confirm the impact of iohexol on GFP, plants expressing GFP were fixed with 2% FA in PBS buffer for 1 h and treated with 20% (w/v) caprylyl sulfobetaine solution for 24 h. The samples were treated with 97% (v/v) TDE (RI = 1.52) or 70.4% (w/w) iohexol solution (RI = 1.52) for 1 h. TDE decreased the GFP fluorescence compared with PBS buffer, whereas iohexol did not affect the GFP fluorescence (Fig. 2d). To assay the effects of iohexol on transparency, completely decolorized

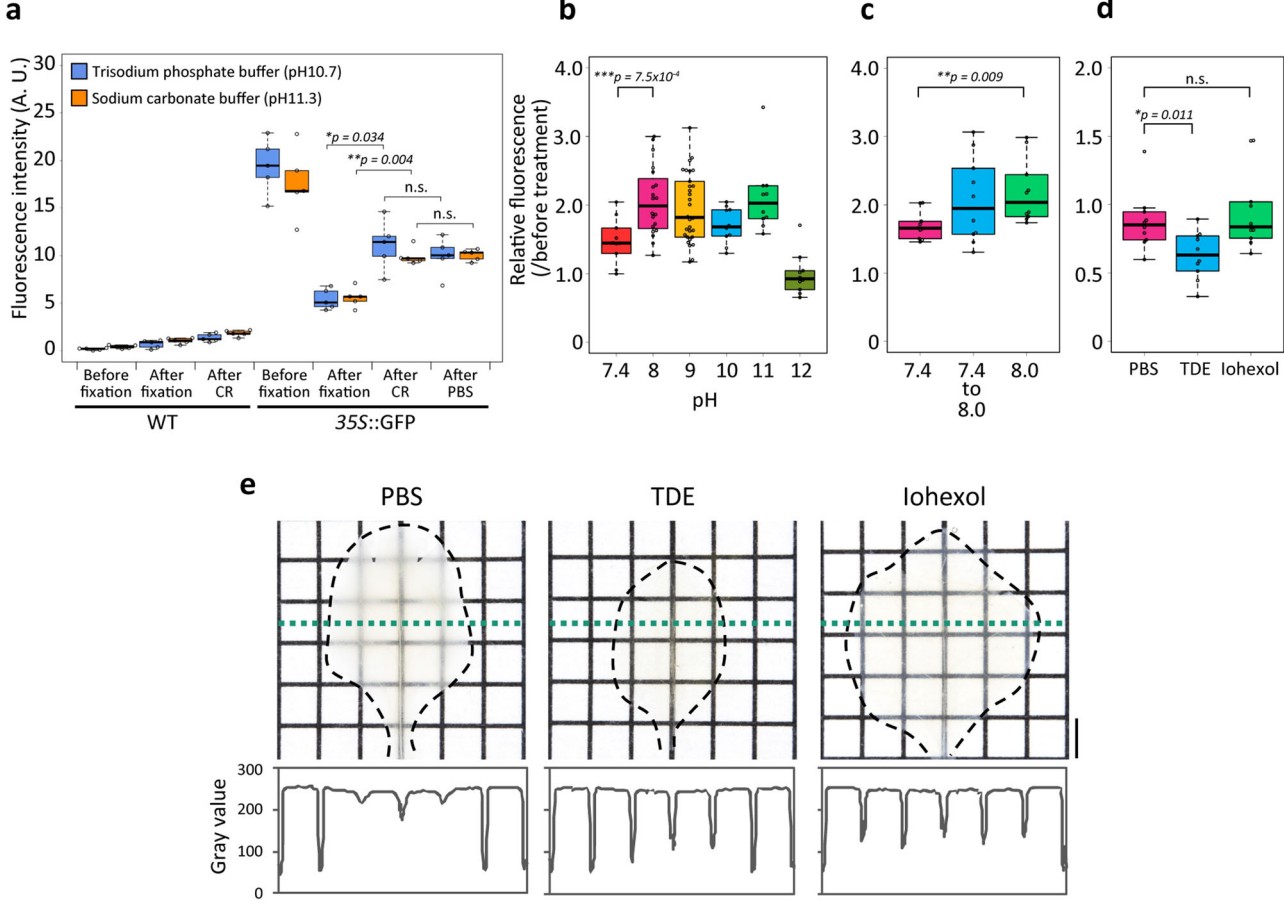

**Fig. 2 Assessment of chemical reactivation and mounting condition. a** Fluorescence intensity was quantified from fluorescence images of the wild-type plants and plants expressing GFP before fixation, after fixation, after alkaline treatments, and after PBS buffer incubation. One-week-old seedlings were fixed with 2% FA and then treated with alkaline solutions for 2 h. Afterward, the seedlings expressing GFP were incubated in PBS buffer at pH 7.4 for 3 h ($n = 5$). **b** Relative fluorescence intensity quantified from fluorescence images of plants expressing GFP before and after incubation in pH 7.4 to pH 12.0 solutions. One-week-old seedlings were fixed with 2% FA and then treated with alkaline solutions for 2 h ($n = 10$ for pH 7.4, 10, 11, and 12; $n = 20$ for pH 8.0; and $n = 30$ for pH 9.0). **c** One-week-old seedlings were fixed with 2% FA and then treated with 20% (w/v) caprylyl sulfobetaine solution at pH 7.4 or 8.0 for 24 h. The seedlings incubated in the pH 7.4 solution were additionally incubated in sodium phosphate buffer pH 8.0 for 2 h ($n = 10$). **d** Relative fluorescence intensity quantified from fluorescence images of plants expressing GFP before and after incubation in mounting reagents ($n = 10$). **e** Decolorized leaves were treated independently with PBS buffer, TDE, and iohexol solutions and placed on a grid. The line graphs indicate the intensity profiles along the green dotted line in the decolorized leaf images. Statistical significance was evaluated using a two-sided Welch's *t*-test. A scale bar = 2 mm.

leaves were fixed with 2% FA and then placed in a 10% (w/v) caprylyl sulfobetaine solution for 3 days. The decolorized leaves were treated with PBS buffer, 97% (v/v) TDE, and 70.4% (w/w) iohexol solutions, and then the samples were placed on grids. The transparency was compared by measuring the intensity of the grids viewed through the leaves (Fig. 2e). Although TDE made the leaves slightly more transparent than iohexol, both TDE and iohexol decreased the intensity of the grids behind the leaves compared with PBS buffer (Fig. 2e). These results suggested that iohexol is a suitable mounting reagent for plant tissues.

**Assessment of fixation conditions to maintain GFP fluorescence.** Finally, we reconsidered the fixation conditions to prevent the reduction in fluorescence of FPs. We investigated the effects of five buffers (PBS, PIPES, HEPES, MOPS, and Tris-HCl) adjusted to pH 7.0 on GFP fluorescence. Seedlings expressing GFP were fixed with 2% FA in each of these buffers for 1 h. PBS and PIPES maintained GFP fluorescence comparatively better than HEPES, MOPS, and Tris-HCl (Fig. 3a). Because the weak alkaline solution was used to recover the GFP fluorescence in the

CR step, the pH of the fixation solution was considered. Three 2% FA in sodium phosphate buffers, at pH 7.0, 7.5, and 8.0, were examined, but they produced similar effects on the GFP fluorescence (Supplementary Fig. 1), which indicated that it was not impacted by the pH of the fixation buffer. In addition, the FA concentration was considered. A 4% FA solution was used for plant tissue fixation; however, it dramatically reduced GFP fluorescence even in PBS buffer (Fig. 3b). Fixation with 1% FA in PBS buffer at pH 7.4 for 1 h more effectively maintained fluorescence, although the fluorescence intensity was reduced to 40% compared with before fixation (Fig. 3b). Insufficient or inappropriate fixation may disrupt or alter cellular structures[22,23]. To examine the effects of 1%-FA fixation on protein localization and cellular structures, the subnuclear localization of a proliferating cell nuclear antigen (PCNA) fused with GFP was observed after fixation. PCNA functions as a sliding clamp to tether DNA polymerase to DNA. It diffuses into the nucleoplasm throughout the cell cycle and produces nuclear speckles during the late S phase in living *A. thaliana* cells (Fig. 3c)[24,25]. Roots of 1-week-old seedlings were incubated with the PBS buffer at pH

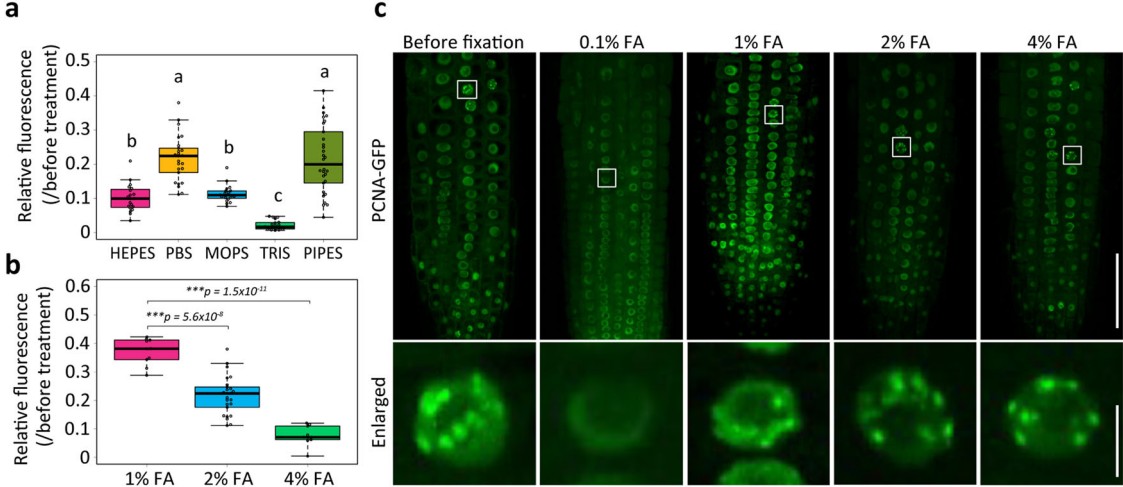

**Fig. 3 Consideration of fixation conditions. a, b** Relative fluorescence intensity was quantified from GFP fluorescence images before and after fixation in HEPES ($n = 20$), PBS ($n = 25$), MOPS ($n = 20$), TRIS ($n = 20$), and PIPES ($n = 30$) buffers supplemented with 2% FA (**a**) and in PBS buffer supplemented with 1% ($n = 10$), 2% ($n = 25$), or 4% FA ($n = 10$) (**b**) for 1 h. Statistical significance was evaluated using the Tukey–Kramer method (**a**) and a two-sided Welch's $t$-test (**b**). The different letters indicate significant differences at $p < 0.05$ (**a**). **c** Confocal micrographs from plants expressing PCNA-GFP fixed in PBS buffer with 0.1, 1, 2, or 4 FA for 1 h captured using the same microscope settings. The enlarged images are indicated by the white open squares in the respective low-magnification images. Scale bars = 50 μm at the upper panel and 5 μm at the lower panel.

7.4 supplemented with 0.1% to 4% FA for 1 h. Nuclear speckles were not detected after treatment with 0.1% FA because the low FA concentration was insufficient to maintain the PCNA localization. The treatments with 1, 2, and 4% FA maintained the subnuclear and speckled localization of PCNA-GFP. Moreover, the signal-to-noise ratio (SNR) was calculated for all the fixation conditions. The fluorescence signals were detected in the nucleoplasm, and the noise was detected in the cytoplasm. The 1% FA produced the highest SNR of all the fixation conditions (Supplementary Fig. 2). Our results suggested that using 1% FA in PBS buffer at pH 7.4 for 1 h is sufficient to fix cellular structures and seems suitable for preserving GFP fluorescence in our clearing method.

**iTOMEI is an outstanding clearing technique for plant organs.** The iTOMEI procedure was compared with the clearing methods ClearSee and TOMEI-II. The total processing times for iTOMEI, ClearSee, and TOMEI-II were 27 h, 4 days, and 3 h, respectively. iTOMEI and ClearSee entirely removed chlorophyll from *A. thaliana* seedlings (Fig. 4a). Seedlings expressing GFP were fixed in PBS buffer supplemented with 1, 2, or 4% FA and subsequently treated with the three clearing methods. iTOMEI maintained the highest fluorescence intensity among the three methods under each fixation condition (Fig. 4b). Unexpectedly, iTOMEI showed comparable GFP fluorescence under all the fixation conditions. This suggested that the GFP quenched by each fixation condition was restored to the same level by the iTOMEI procedure; consequently, when performing iTOMEI, a suitable FA concentration for fixation can be selected.

After applying the three clearing techniques, confocal fluorescence images of leaves expressing histone H2B-GFP were captured (Fig. 4c). One leaf was fixed with 4% FA and cut into four pieces. An individual piece was treated with PBS buffer or each clearing method. The fluorescent signals were almost invisible at 30 μm in the leaf piece treated with PBS buffer or TOMEI-II. Although GFP signals were detected at 30 μm in the leaf piece cleared using ClearSee or iTOMEI, the brightest fluorescence at 30 μm was detected after the iTOMEI treatment (Fig. 4c). The images were captured from the adaxial to the abaxial side of each leaf piece. The GFP signals in the nuclei of stomatal cells on the abaxial epidermis

were measured to compare the fluorescence intensities between the clearing methods, and the background noise was measured from the cytoplasm of the abaxial epidermis to calculate SNRs. GFP signals were not detected in the PBS-treated leaf piece. iTOMEI showed the highest GFP fluorescence intensity and SNR of the three methods (Fig. 4d, e). These results strongly suggest that iTOMEI can be used to better visualize fluorescent proteins in plant tissues than conventional methods.

The iTOMEI protocol is as follows: Step 1: A sample is fixed with 1, 2, or 4% FA in PBS buffer for 1 h. Step 2: The fixed sample is decolorized in sodium phosphate buffer at pH 8.0 supplemented with 20% (w/v) caprylyl sulfobetaine for 24 h. Step 3: The sample is sequentially incubated in 20, 50, and 70.4% (w/w) iohexol solution for 10 min each and then in 70.4% (w/w) iohexol solution (RI = 1.52) for 1 h.

**iTOMEI revealed FP expression patterns inside plant tissues and organs.** Next, we performed iTOMEI to clear organs of rice (*Oryza sativa*). Three days were required for the young leaf to become completely transparent (Fig. 5a). We also observed the expression pattern of the transcription factor OsMADS15, which regulates the transition from the vegetative to the reproductive phase in the shoot apical meristem (SAM). Because the diameter of the rice reproductive SAM is ~150 μm, the FP expression pattern in the rice SAM had not been analyzed previously without sectioning. We attempted to observe FPs in the SAM using iTOMEI without sectioning. After 4%-FA fixation of the SAM dissected from plants expressing OsMADS15 fused with mOrange[26] (Supplementary Fig. 3), the SAM was stained with a fluorescent cell-wall-staining optical brightener, SCRI Renaissance 2200 (SR2200), and cleared with TOMEI-II or iTOMEI. Finally, the SAM was incubated with 70% (w/v) TDE (RI = 1.47) and 56.2% (w/w) iohexol solution (RI = 1.46) for TOMEI-II and iTOMEI, respectively, to adjust the RIs of the samples to that of glycerol. Neither fluorescent signal was detected in the central area of the SAM in only fixed SAMs (Fig. 5b). The SR2200 signal was detected in the central portion of the SAM in TOMEI-II-treated samples, whereas the nuclear-specific OsMADS15-mOrange signal was not detected at the 80-μm depth. After clearing with iTOMEI, both signals were detected in the central region of

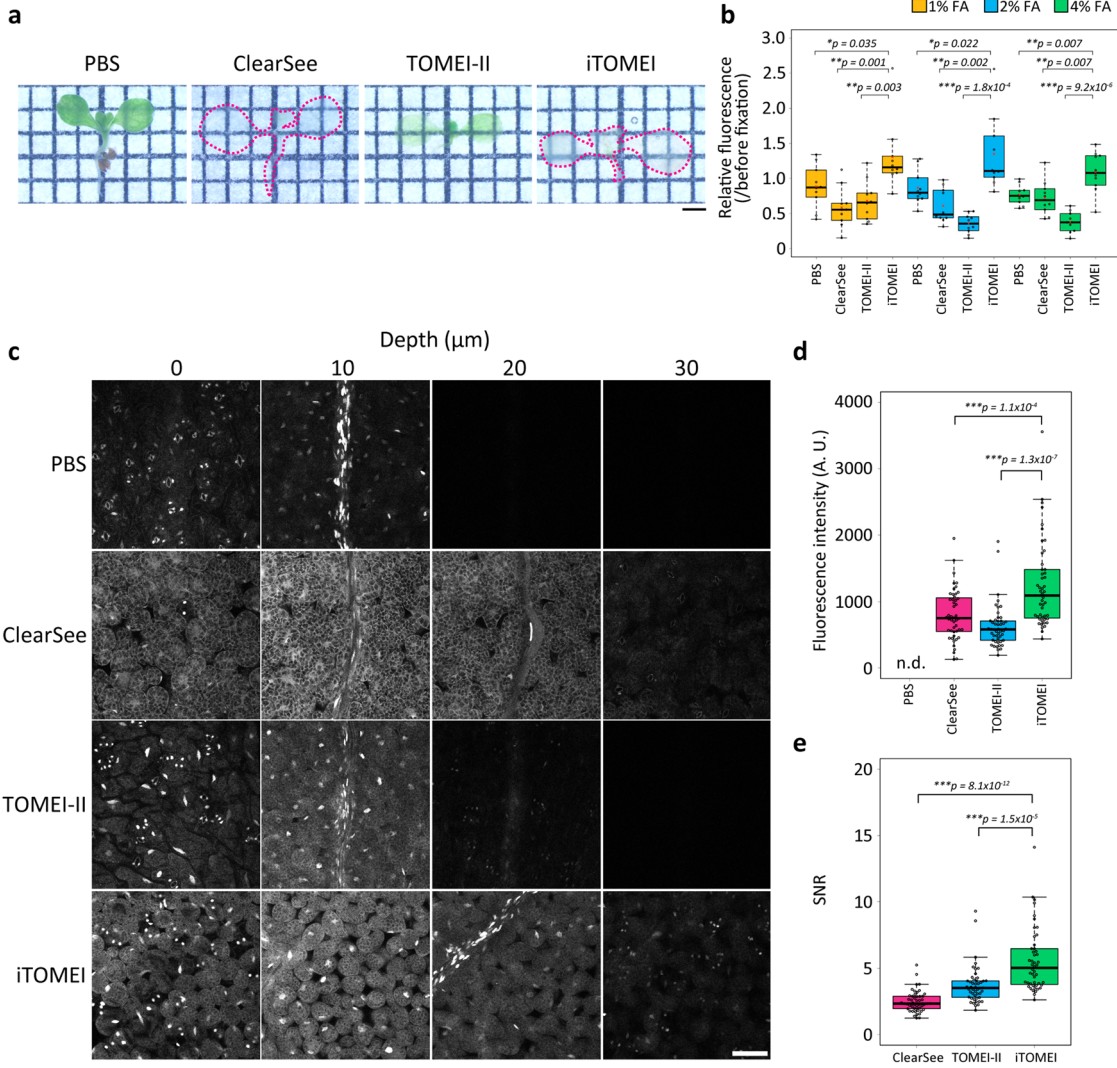

**Fig. 4 Comparison of clearing techniques for plant organs. a** Seedlings treated with PBS buffer, ClearSee, TOMEI-II, and iTOMEI. **b** Relative fluorescence intensity quantified from fluorescence images of plants expressing GFP before and after clearing methods. One-week-old seedlings were fixed with 1, 2, and 4% FA and then treated with PBS buffer and three methods. $n = 10$ plants. **c** Leaf of a plant expressing histone H2B-GFP treated with PBS buffer and the three clearing methods. Confocal optical sections were captured from the adaxial to the abaxial surface under the same optical conditions. **d** Fluorescence intensity of H2B-GFP in the nucleus of stomatal cells on the abaxial epidermis. The intensity was measured after PBS buffer and three clearing treatments. GFP signals were not detected in the PBS-treated leaf piece. $n = 48$ nuclei. **e** SNR was calculated by H2B-GFP signals in the nucleus and background noise in the cytoplasm. $n = 48$ nuclei. Statistical significance was evaluated using the two-sided Welch's $t$-test. Scale bars $= 1$ mm (**a**) and $50\,\mu$m (**c**).

the SAM at the 80-μm depth. OsMADS15-mOrange was strongly expressed in the basal region of a hairy bract and in the outer two cell layers of an incipient primary branch meristem (Fig. 4b), suggesting that iTOMEI can reveal the expression pattern of FPs inside the larger reproductive SAM. We also attempted to detect auxin signaling in the root tips of plants expressing *pDR5rev*::NLS-3xVenus. iTOMEI produced the brightest *pDR5rev*::NLS-3xVenus signals in the root tips compared with other methods (Fig. 5c). The *pDR5rev*::NLS-3xVenus construct was strongly expressed in the central and surrounding metaxylem cells in the stele, quiescent center, and columella cells. In an optical cross-section of the root, five FP foci surrounding the signal in the center of the root were detected, corresponding to the central and surrounding metaxylem (Fig. 5d).

Next, we attempted to clear organs of the liverwort *Marchantia polymorpha*. An apical portion of a mature thallus of *M. polymorpha* was cleared within 27 h, as for *A. thaliana* seedlings (Fig. 6a). Calcofluor White M2R and histone H2B-tdTomato signals were observed from the ventral to the dorsal surfaces of

the thallus in a 3-day-old gemmaling (Fig. 6b and Supplementary Fig. 4). In the apical region of the thallus, we detected an apical cell or subapical cells that appeared fan-shaped when viewed in the Y–Z optical section (Fig. 6b)[27]. Thalli of a 2-week-old gemmaling expressing H2B-tdTomato were fixed and treated with PBS buffer, TOMEI-II, and iTOMEI. tdTomato signals were observed from the dorsal to the ventral surfaces in the thalli using confocal microscopy. Fluorescent signals were detected at a 200-μm depth in the iTOMEI-treated gemmaling but not in PBS- and TOMEI-II-treated gemmalings (Fig. 6c). The signal intensity of tdTomato was measured in confocal optical sections at 100- to 110-μm depths after the three treatments. iTOMEI produced the highest signal intensity (Fig. 6d). These observations demonstrated that iTOMEI is suitable for the deep imaging of *M. polymorpha* thalli.

*RSL* class I genes encode basic helix-loop-helix transcription factors conserved among land plants. These transcription factors positively regulate root hair formation in *O. sativa* and *A. thaliana* and rhizoid development in *M. polymorpha* and

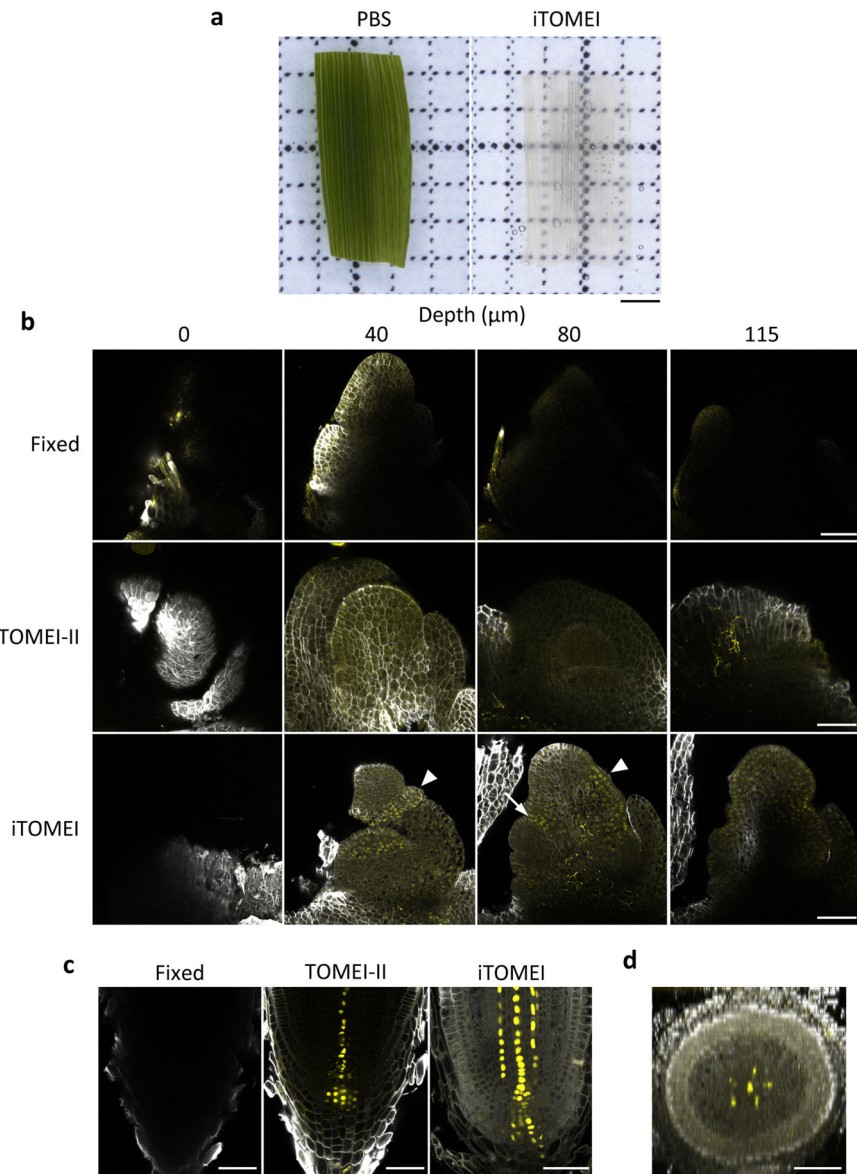

**Fig. 5 Deep imaging of transparent rice organs using iTOMEI. a** Leaf of 5-day-old seedling of *O. sativa* treated with PBS buffer and iTOMEI. **b** Shoot apical meristem of a plant expressing MADS15-mOrange (yellow) stained with SR2200 (white) and treated with PBS buffer, TOMEI-II, and iTOMEI. Confocal optical sections were captured under the same optical conditions. The arrow indicates a strong mOrange signal in the basal region of a hairy bract. The arrowhead indicates mOrange signal at the incipient primary branch meristem. **c, d** Root tip of a plant expressing DR5rev::NLS-3xVenus (yellow) stained with SR2200 (white) and treated with PBS buffer, TOMEI-II, and iTOMEI. Confocal longitudinal (**c**) and transverse optical sections (**d**) of the root are shown. Scale bars = 1 mm (**a**) and 50 μm (**b–d**).

*Physcomitrium patens*[28–31]. *RSL* class I genes are expressed in a precursor cell of the root hair and rhizoid in *O. sativa*, *A. thaliana*, and *P. patens*. However, the expression pattern of an ortholog of *M. polymorpha*, Mp*RSL1*, has not been investigated. The expression pattern of nuclear tdTomato under control of the Mp*RSL1* promoter in the gemma showed that the promoter was activated in rhizoid precursor cells as expected and, unexpectedly, in the apical regions (Fig. 6e). Thus, Mp*RSL1* may function in the apical regions of gemmae. Next, the gemma cup, a cup-shaped receptacle on the mature thallus, was cleared using iTOMEI. Using a two-photon excitation microscope, the gemmae were observed within the transparent gemma cup through the wall of the gemma cup (Fig. 6f). Many immature gemmae are present in a gemma cup and develop asynchronously, which enables observations of gemmae at various developmental stages within a gemma cup (Fig. 6g). Mp*RSL1* was strongly expressed in the

preapical regions of the immature gemmae at different developmental stages (Fig. 6g, arrowheads). Additionally, Mp*RSL1* expression was detected in rhizoid precursor cells in large immature gemmae (Fig. 6g, arrows) but not in smaller gemmae. These results demonstrate that iTOMEI in combination with a two-photon excitation microscope enables the in situ observation of gene expression patterns during gemma development through the wall tissues of the gemma cup.

**iTOMEI contributes to deep imaging of the mouse brain**. We tested whether the application range of iTOMEI could be expanded to animal tissues. We modified the iTOMEI protocol used for the mouse brain by referencing the SeeDB2 protocol and designated the procedure iTOMEI-Brain (iTOMEI-B). A fixed mouse brain was sectioned into 2-mm slices and treated with PBS

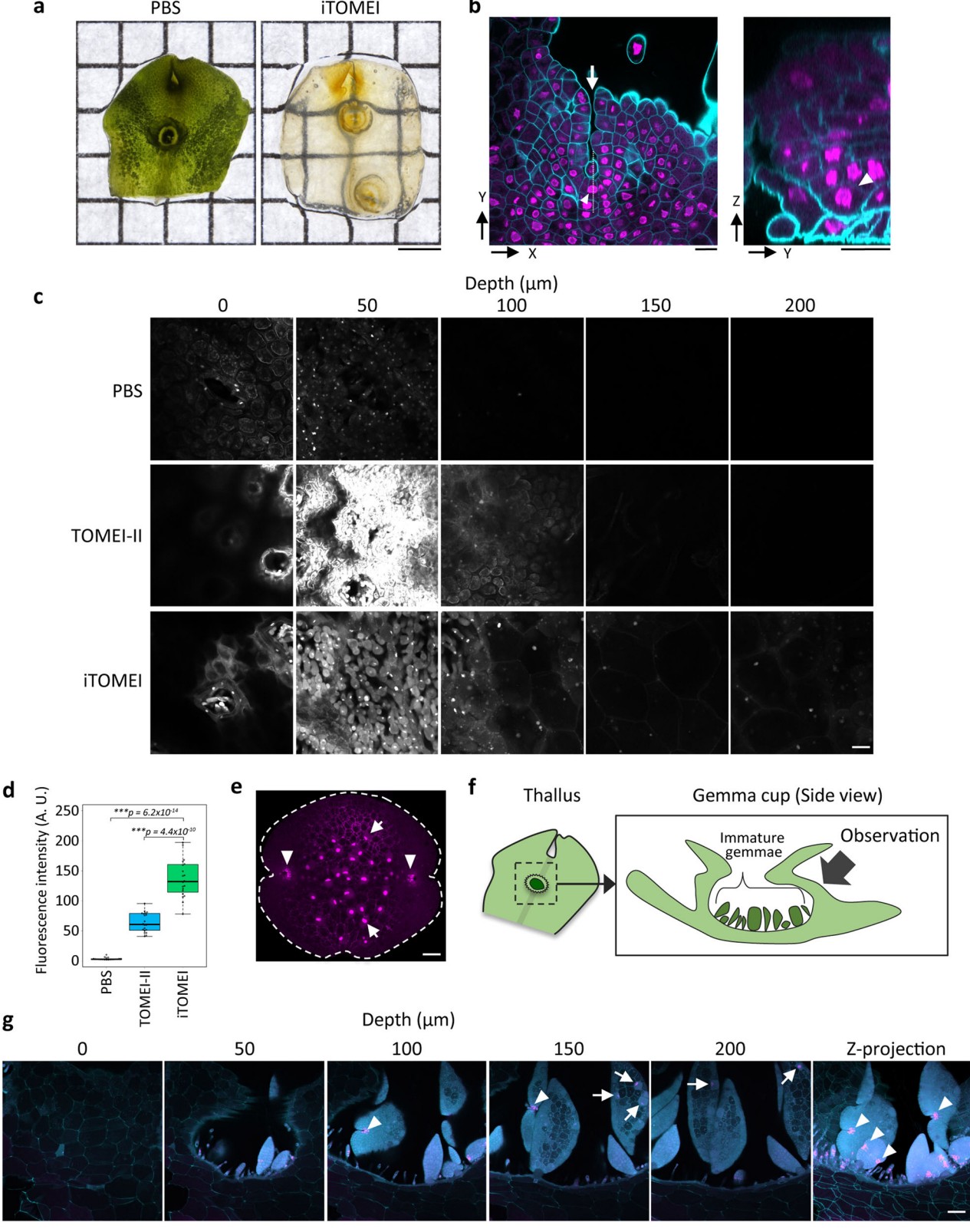

buffer, SeeDB2, and iTOMEI-B. Although a 24-h treatment with iTOMEI-B was insufficient to clear the brain, a 48-h treatment achieved high transparency (Supplementary Fig. 5a). iTOMEI-B inhibited the change in slice size to the same degree as SeeDB2 (Supplementary Fig. 5a). Hemi brains expressing EGFP were also treated with PBS buffer, SeeDB2, and iTOMEI-B and observed from the brain's cortical surface using a two-photon excitation microscope. iTOMEI-B allowed the acquisition of fluorescence images to a 3-mm depth in a transparent brain expressing GFP (Supplementary Fig. 5b). The GFP fluorescence was clearly detected in the somata, axons, and dendrites of the hippocampus and cerebral cortex (Supplementary Fig. 5b–d). Surprisingly, an axonal projection was observed in the thalamus at the depth of 3 mm from the cortical surface. Deep imaging of the transparent

**Fig. 6 Deep imaging of transparent organs of *M. polymorpha* using iTOMEI. a** Thallus of *M. polymorpha* treated with PBS buffer and iTOMEI. **b** Thallus of a plant expressing histone H2B-tdTomato (magenta) fixed with 1% FA and then treated with iTOMEI and stained with Calcofluor White (cyan). Confocal optical sections were captured from the ventral to the dorsal surfaces of the apical region and were reconstructed in the Y–Z plane image at a dotted line in the X–Y plane image. Arrowheads indicate a fan-shaped cell, and an arrow indicates an apical notch. **c** Thalli of 2-week-old gemmalings expressing histone H2B-tdTomato treated with PBS buffer, TOMEI-II, and iTOMEI. Confocal optical sections were captured from the dorsal surface under the same optical conditions. **d** Fluorescence intensity of histone H2B-tdTomato was measured in a confocal optical section at 100- to 110-μm depths ($n = 20$). Statistical significance was evaluated using a two-sided Welch's *t*-test. **e** Gemma expressing *pMpRSL1::tdTomato-NLS* (magenta) treated with iTOMEI. Arrowheads indicate meristematic cells showing high fluorescent signals. The broken line indicates the gemma margin. The arrow indicates a rhizoid precursor cell. **f** Schematic illustration of a thallus and a gemma cup. When observed with a two-photon excitation microscope, the gemmae were observed through the sidewall of the gemma cup. **g** Gemma cup of a plant expressing *pMpRSL1::tdTomato-NLS* (magenta) stained with Calcofluor White (cyan) and treated with iTOMEI. The optical sections were captured with a two-photon excitation microscope. The Z-projection image was created from optical sections between the depths of 60 and 120 μm. Scale bars = 2 mm (**a**), 20 μm (**c**), and 50 μm (**b**, **e**, **g**).

brain using iTOMEI-B enabled the identification of the layered structures of the hippocampus and cortex. The characteristic dendritic morphology of CA1 pyramidal neurons was maintained (Supplementary Fig. 5d). Our imaging data suggested that iTOMEI can be adapted to not only plant tissues but also the mouse brain.

## Discussion

The iTOMEI procedure can produce bright FP fluorescence of increased intensity in transparent organs and tissues compared with conventional methods. This enables the three-dimensional detection of FP signals from a 200-μm depth in plant tissues. We have adopted four methods to make plant tissues transparent without quenching FPs: Sca*l*e-based method, ClearSee, PEA-CLARITY, and TOMEI-II. iTOMEI has an advantage over the other methods, except TOMEI-II, in that the process can be completed in as little as 27 h. Compared with ClearSee and TOMEI-II, which have relatively short processing times, iTOMEI has the advantage of detecting the greatest GFP fluorescence intensity in deep tissues. In addition, iTOMEI produces a higher SNR than ClearSee and TOMEI-II, which enables fluorescence observations with low background noises. It can be used with multiple plant species, similar to other transparency methods. Thus, iTOMEI is a useful short processing-time transparency method.

Chemical reactivation enhances the fluorescence intensities of GFP and YFP in resin-embedded biological tissues[20,21]. Alkaline solutions, in the range from pH 9.0 to 12.0, are generally used for CR. Because biological tissues are morphologically damaged by solutions having such high pH levels, the adaptation of CR for sample preparation without embedding in resin seemed difficult. However, we showed that a weak alkaline solution of pH 8.0 harbored the fluorescence recovery capability (Fig. 2b). In addition, PBS buffer at pH 7.4 partly reactivated GFP. These results suggested that sample incubation in solutions at pH 7.4 to 8.0 can reactivate FPs quenched by fixation.

Fixation greatly reduces the fluorescence intensity of GFP in *A. thaliana*. The fluorescence intensity of a sample fixed with 1% FA was approximately two- and four-times higher than those of samples fixed with 2% and 4% FA, respectively (Fig. 3b). Surprisingly, the fluorescence intensity of the sample after the iTOMEI processing was not greatly affected by FA concentration. The fluorescence intensities of iTOMEI-treated samples fixed with 1% FA were less than double those of iTOMEI-treated samples fixed with 4% FA (Fig. 4b). PBS-treated samples also showed a similar trend, suggesting that GFP inactivated by high FA concentrations was easily recovered during each procedure because incubation in near-neutral solutions can recover GFP fluorescence.

The RI of the mounting solution containing 70.4% (w/w) iohexol for iTOMEI was adjusted to that of immersion oil (RI = 1.52). When a glycerol immersion objective was used, the RI of the PBS-diluted mounting solution (RI = 1.46) containing 56.2% (w/w)

iohexol was adjusted to that of glycerol. When mounting solutions having different RIs from immersion solutions were used, spherical aberrations were enhanced, and the image resolution decreased. Thus, when taking high-resolution images, the RI of the mounting solution should be properly adjusted.

The clearing of the rice SAM by iTOMEI allowed the spatial expression-profiling of the key transcription factor OsMADS15 at a three-dimensional single-cell resolution (Fig. 5b–d). Os*MADS15*, which is a homolog of *APETALA1/FRUITFUL* in *A. thaliana*, is essential for bract and floral organ development in rice[32]. Consistent with the reported function, the OsMADS15-mOrange signal was observed at the incipient primary branch meristem, which provides flowers at an advanced stage, and in the basal region of a hairy bract, which is a suppressed leaf produced by the reproductive SAM[33]. Os*MADS15* expression is regulated by the florigen Hd3a in the rice SAM[26,34]. Thus, our results suggested the action site of rice florigen at a single-cell resolution.

In the Mp*rsl1-1* mutant of *M. polymorpha*, rhizoids did not develop from the ventral surfaces of the thalli, gemmae and mucilage papillae did not develop in gemma cups, and slime papillae did not develop near the apical region[30]. However, whether the Mp*rsl1-1* mutant shows any defects in the apical region has not been determined. We revealed that Mp*RSL1* promoter activity was detected in rhizoid precursor cells and the apical regions in mature gemmae, and it was restricted in the apical region of the gemma cup during gemma development (Fig. 6g). This imaging data suggested that Mp*RSL1* is a useful marker to detect meristematic cells in liverworts.

Although iTOMEI was developed for plant tissues and organs, it can be applied to the mouse brain (Supplementary Fig. 5). Because each clearing method has its own characteristics, the method should be used properly depending on in the targets of the imaging analyses. iTOMEI-B achieved high transparency and high GFP detection in the brain compared with SeeDB2, and it revealed minor alterations in brain volume compared with those observed using SeeDB2 (Supplementary Fig. 5a). The high transparency produced by iTOMEI-B potentially contributed to observations of weak FP expression in the cells located deep in the brain.

Recent developments in transparent techniques enable us to analyze the expression patterns of FPs in whole tissues and organs. Our developed iTOMEI is a powerful technique to transparentize plant organs with almost no attenuation of the fluorescence from FPs, which were expressed at the 200-μm depth in organs. It will also be a helpful technique to construct an image platform of organ morphology at a single-cell resolution.

## Methods

**Plant materials and growth conditions**. *Arabidopsis thaliana* ecotype Columbia-0 was used as the wild type. The transgenic lines *promoter35S*::GFP, *pPCNA1*::PCNA1-EGFP, and *p35S*::histone H2B-GFP were previously

reported[25,35,36]. Seeds were germinated and grown on half-strength Murashige and Skoog plates supplemented with 1% sucrose and 14-day-old plants were transferred to soil. Plants were grown in a growth chamber (16 h light/8 h dark, 22 °C).

*Oryza sativa* 'Nipponbare' was used for shoot apical meristem (SAM) analysis. Plants were grown in the growth chamber under short-day conditions (10 h light at 27 °C and 14 h dark at 25 °C, light intensity 400–700 nm, 100 µmol m$^{-2}$ s$^{-1}$). Vegetative and reproductive SAMs were isolated by hand dissection of the basal region of rice under a microscope[37]. Generation of the transgenic lines *OsMADS15-mOragene* and *DR5rev:NLS-3xVenus* was previously reported[26,38].

A male accession of *M. polymorpha* Takaragaike-1 (Tak-1) was used as the wild type. Plants were grown on half-strength Gamborg's B5 plates in a growth chamber (16 h light/8 h dark, 22 °C).

**Plasmid construction and transformation.** The Mp*RSL1* (Mp3g17930) genomic region, including a 6964-bp fragment upstream of the 15th Met codon in the first exon, was amplified as the promoter region from Tak-1 genomic DNA by PCR using a gene-specific primer pair (pMpRSL1-F1, CACCCCCAAATGCAATTCTA TTGTGTATTCAT; pMpRSL1-R, AGCTGGGTCGGCGCGCATGTTGTTCCTCC TGCTCAGTGT) and subcloned into the pENTR/D-TOPO vector (Thermo Fisher Scientific). A DNA cassette including the promoter region was transferred to a binary vector pMpGWB116[39] using the Gateway LR Clonase II Enzyme mix (Thermo Fisher Scientific). To create a construct for *pMpEF1::H2B-tdTomato*, a SalI-NotI fragment including a coding sequence for H2B-tdTomato from the plasmid spUC-RPS5A::H2B-tdTomato[40] was ligated with SalI- and NotI-digested pENTR-1A vector (Thermo Fisher Scientific), and the resulting vector was used for LR reaction with a binary vector pMpGWB303[39]. The binary vectors were used for *Agrobacterium*-mediated transformation of Tak-1 thalli and Tak-1 × Tak-2 F1 sporelings, respectively[41].

**Measurements of fluorescence intensity in *A. thaliana*.** One-week-old seedlings expressing GFP were fixed with 2% FA (Polyscience) in PBS buffer for 1 h. The purchased FA solution comes sealed in an ampoule, and the ampoule was opened just before use. The PBS buffer (137 mmol/l NaCl, 8.1 mmol/l Na$_2$HPO$_4$, 2.68 mmol/l KCl, and 1.47 mmol/l KH$_2$PO$_4$, pH 7.4) was prepared in the laboratory. The fluorescent images of seedlings were captured under the same optical conditions using a fluorescence stereomicroscope (SMZ18; Nikon) equipped with a DS-Ri2 digital camera (Nikon) in Figs. 1b, g, 2b, d, and 3a, b or a fluorescence stereomicroscope (SZX16; Olympus) equipped with a DP21 camera (Olympus) only in Fig. 2c and 4b. To measure the fluorescence intensity, the same areas of the hypocotyls in the same plants before and after treatment were selected as ROIs, and an area with no plants was selected as background in each image using ImageJ 1.53k. The fluorescence intensity was calculated as the value of the fluorescent signal in the ROI minus that in the background.

To calculate the SNRs of PCNA signals in confocal images, we randomly selected five pixels each from a nucleus that showed no speckled structures and from cytoplasm in confocal optical sections. The SNR was calculated by averaging the fluorescence intensity of five pixels in the nucleus and dividing by the pixel intensity in the cytoplasm. To calculate the fluorescence intensity of H2B-GFP in *A. thaliana*, we selected the nuclei of stomatal cells on the abaxial epidermis as the fluorescent signals and cytoplasm as the background in confocal optical sections. In the case of H2B-tdTomato in *M. polymorpha*, we selected a fluorescent signal from the nucleus and background noise from the cytoplasm in confocal optical sections at 100- and 110-µm depths. The fluorescent intensity was defined as the value of the intensity of the nucleus minus that of the background. An SNR was defined as the value of the intensity of the background over that of the nucleus.

**Preparation of the alkaline solutions.** We prepared the 0.5 mM trisodium phosphate buffer at pH 10.7 and the 100 mM sodium carbonate buffer at pH 11.3. We modified the pH of sodium phosphate buffer to prepare buffers at pH 8.0 and 9.0 by mixing 100 mM disodium hydrogen phosphate solution and 100 mM sodium dihydrogen phosphate solution as a ratio of 95:5 and 100:0, respectively, and 50 mM trisodium phosphate solution and water to reach pH 10, 11, and 12. pH of all buffers was measured by a pH meter (Horiba).

**Measurements of chlorophyll absorbance.** Two-week-old plants were fixed with 2% FA in PBS buffer for 1 h. A fixed seedling was placed into a plastic tube containing water supplied with 10% (w/v) various detergent and was incubated for 24 h in the dark (Supplementary Table 1). As a negative control, the detergent solutions without the seedling were incubated for 24 h. The absorbance of chlorophylls, eluted from the seedling to the detergent solution, was measured at 674 nm using an absorption spectrophotometer (Nanophotometer Pearl; IMPEL). For the determination of the caprylyl sulfobetaine concentration, 20-day-old plants were used to measure chlorophyll absorbance (Fig. 1a).

**Optical clearing of *A. thaliana* by iTOMEI.** Seedlings and leaves were fixed in PBS buffer with 1, 2, and 4% FA for 1 h with evacuation for the first 10 min and then the samples were washed three times in PBS buffer for 5 min. The fixed samples were treated with a decolorization solution (100 mM sodium phosphate buffer [a mixture of 100 mM disodium hydrogen phosphate and 100 mM sodium dihydrogen

phosphate at 95: 5] at pH 8.0 supplemented with 20% (w/v) caprylyl sulfobetaine [TCI]) for 24 h with gentle shaking in the dark. After washing in PBS buffer for 5 min, the samples were incubated in serial mounting solutions (20, 50, and 70.4% (w/v) iohexol [TCI or Merck] in PBS buffer) for 10 min each to avoid drastic osmotic changes and in 70.4% (w/v) iohexol for 1 h with gentle shaking in the dark. Finally, the samples were mounted on a glass slide in the mounting solution containing 70.4% (w/v) iohexol and a coverslip applied. Exposure of the mounting solution to air for several minutes results in water evaporation and a skin form on the solution surface. To inhibit skin formation, the mounting procedure should be conducted rapidly. As the mounting solution is prone to mold growth, it should be stored at 4 °C. All procedures were performed at 25 °C.

**Optical clearing of *A. thaliana* by TOMEI-II.** Seedlings and leaves were fixed in PBS buffer with 1, 2, and 4% FA for 1 h with evacuation for the first 10 min and then the samples were washed three times in PBS buffer for 5 min. The fixed samples were gradually treated in PBS buffer with a graded series of Tissue-Clearing Reagent TOMEI (97% (v/v) TDE and 0.0025% (w/v) propyl gallate in PBS buffer; TCI) for 10 min each (10, 30, 50, and 70% (v/v)) in the dark. Subsequently, the samples were incubated in 100% (v/v) TOMEI or 80% (v/v) TOMEI for 1 h, as shown in Fig. 4b, c, respectively. Finally, the samples were mounted between the glass slide and the coverslip. All procedures were performed at 25 °C.

**Optical clearing of *A. thaliana* by ClearSee.** Seedlings and leaves were fixed in PBS buffer with 1, 2, and 4% FA for 1 h with evacuation for the first 10 min and then the samples were washed three times in PBS buffer for 5 min. The fixed samples were treated in ClearSee solution [10% (w/v) xylitol, 15% (w/v) sodium deoxycholate, 25% (w/v) urea; Fujifilm] for 4 days in the dark. Finally, the samples were mounted between the glass slide and the coverslip using ClearSee solution. All procedures were performed at 25 °C.

**Optical clearing of *O. sativa* and *M. polymorpha* by iTOMEI.** A 5-day-old leaf of rice plant was fixed with 4% FA for 1 h with evacuation for the first 10 min and the samples were washed three times in PBS buffer for 5 min. The fixed sample was incubated in the decolorization solution for 3 days with gentle shaking. The solution was exchanged with the mounting solution containing 56.2 (w/w) iohexol (RI = 1.46) and the samples were incubated for 1 h with gentle shaking. The SAMs and roots of rice plants were isolated by hand dissection of wild-type and transgenic plants (Supplementary Fig. 3). The isolated samples were fixed in PBS buffer with 4% FA for 1 h with evacuation for the first 10 min. The samples were stained in 0.1% (v/v) SR2200 (solution from supplier was considered as 100%: Renaissance Chemicals) at this step. Then the samples were washed three times in PBS buffer for 5 min for the first washing and 10 min for the second and third washing. The fixed samples were treated in the decolorization solution for 24 h with gentle shaking. The samples were incubated in the mounting solution containing 56.2 (w/w) iohexol for 1 h without shaking. Finally, the samples were mounted between the glass slide and the coverslip using the mounting solution.

Thalli, gemmalings, and gemmae of *M. polymorpha* were fixed in PBS buffer supplemented with 1% FA for 1 h with evacuation for the first 10 min, and then the samples were washed three times in PBS buffer for 5 min each. The fixed samples were treated in the decolorization solution for 24 h with gentle shaking in the dark. After washing in PBS buffer for 5 min, the samples were stained in Calcofluor White Stain (1 g/l Calcofluor White M2R and 0.5 g/l Evans blue; Merck) for 10 min. After washing in PBS buffer for 5 min, the samples were incubated in the mounting solution containing 70.4 (w/w) iohexol for 1 h with gentle shaking in the dark. Finally, the samples were mounted between the glass slide and the coverslip using the mounting solution. For observation of immature gemmae in gemma cups using a two-photon excitation microscope, the whole gemma cup was mounted and the immature gemmae were observed through the sidewall of the gemma cup (Fig. 6f, g). All procedures were performed at 25 °C.

**Optical clearing of *O. sativa* and *M. polymorpha* by TOMEI-II.** The SAMs and roots of rice plants were isolated by hand dissection of wild-type and transgenic plants. The isolated samples were fixed in PBS buffer with 4% FA for 1 h with evacuation for the first 10 min. The samples were stained in 0.1 % (v/v) SR2200 (solution from supplier was considered as 100%: Renaissance Chemicals) at this step. Then the samples were washed three times in PBS buffer for 5 min for the first washing and 10 min for the second and third washing. The samples ware incubated in 10, 30, 50, and 70% (v/v) (RI = 1.47) TOMEI-II solution for 10 min sequentially. After clearing, the samples were mounted in 70% (v/v) TOMEI-II.

Two-week-old gemmalings were fixed in PBS buffer with 4% FA for 1 h with evacuation for the first 10 min and then the samples were washed three times in PBS buffer for 5 min. The fixed samples were gradually treated in PBS buffer with a graded series of Tissue-Clearing Reagent TOMEI (TCI) for 10 min each (10, 30, 50, 70, and 100% (v/v)) in the dark. The samples were incubated in 100% (v/v) Tissue-Clearing Reagent TOMEI for 1 h with gentle shaking in the dark. Finally, the samples were mounted between the glass slide and the coverslip using 100% (v/v) Tissue-Clearing Reagent TOMEI. All procedures were performed at 25 °C.

**Mice and surgery**. All experimental protocols were evaluated and approved by the Regulation for Animal Research at Tokyo University of Science. All experiments were conducted in accordance with the Regulations for Animal Research at the Tokyo University Science. Adult C57Bl/6 J male (4–5 months old) mice were used. Mice were maintained under a 12 h light/12 h dark cycle (light period 07:30–19:30), and ad libitum feeding and drinking conditions. The plasmid AAV8 CaMKIIa-EGFP was purchased from UNC Vector Core. We used a titer of approximately $1 \times 10^{12}$ vg/ml of EGFP viruses in this study. Mice were mounted in a stereotaxic apparatus, anesthetized with pentobarbital (80 mg/kg) and subcutaneously injected carprofen (5 mg/kg) and dexamethasone (0.2 mg/kg). A 2-mm-diameter craniotomy was performed above the hippocampus. A 0.3 μl virus solution was infused using a Hamilton syringe through a glass micropipette at the following coordinates: relative to bregma (mm): anteroposterior axis (AP):−2.0, mediolateral axis (ML): 1.4, and dorsoventral axis (DV): 0.6, 1.2, and 1.7 from dura mater, taken from the mouse brain atlas[42] at a rate of 0.1 μl/min. A glass capillary was left in place for an additional 10 min. A brain sample was harvested 4 weeks after surgery to allow for recovery and sufficient expression of genes.

**Mouse brain sample preparation**. Mice were deeply anesthetized with pentobarbital and transcardially perfused with 4% paraformaldehyde in 0.1 M sodium phosphate buffer (pH 7.4). The brains were excised, sagittally dissected, and postfixed with the same fixative at 4 °C overnight. The fixed brains were sectioned at a thickness of 2 mm with a laser blade to make brain slices.

**Optical clearing of the mouse brain by iTOMEI-B**. The hemi brain and brain slices were treated with 20% (w/v) caprylyl sulfobetaine in 0.1 M sodium phosphate buffer (pH 8.0) for 16 h. The samples were next treated with 18.7% (w/w) iohexol and 20% (w/v) caprylyl sulfobetaine in PBS buffer for 6 h. Subsequently, the samples were treated with 28.1% (w/w) iohexol and 20% (w/v) caprylyl sulfobetaine in PBS buffer for 6 h and then with 56.2% (w/w) iohexol in PBS buffer for 12 h. Finally, the hemi brains were treated with 56.2% (w/w) iohexol in PBS buffer for 8 h and the brain slices were treated with 70.4% (w/w) iohexol in PBS buffer for 8 h. The hemi brains were mounted with Omnipaque 350 and observed by a two-photon excitation microscope. The brain slices were mounted with 70.4% (w/w) iohexol in PBS buffer and observed with a stereomicroscope. All procedures were performed at 25 °C.

**Optical clearing of the mouse brain by SeeDB2**. The hemi brain and brain slices were treated with 2% saponin in PBS buffer for 16 h. The samples were next treated with 2% saponin in a 1:2 mixture of Omnipaque 350 (Daiichi-Sankyo) and water for 6 h. Subsequently, the samples were treated with 2% saponin in a 1:1 mixture of Omnipaque 350 and water for 6 h and then 2% saponin in Omnipaque 350 for 12 h. Finally, the hemi brains were treated with Omnipaque 350 for 8 h and the brain slices were treated with 2% saponin in SeeDB2S for 8 h. The hemi brains were mounted in Omnipaque 350 and observed with a two-photon excitation microscope. The brain slices were mounted with SeeDB2S and observed with a stereomicroscope. All procedures were performed at 25 °C.

**Confocal microscopy and two-photon excitation microscopy**. The confocal images of *A. thaliana* and *M. polymorpha* were obtained with a confocal laser-scanning microscope (FV1000 and FV1200; Olympus) equipped with 405, 473, and 559 nm LD laser lines and a 60× 1.40 N.A. oil immersion objective (PlanApo; Olympus), a 40× 1.30 N.A. oil immersion objective (UPlanFL; Olympus), and a 20× 0.75 N.A. dry objective (UPlanSApo; Olympus). Two-week-old gemmalings of *M. polymorpha* were observed with a confocal laser-scanning microscope (LSM710; Carl Zeiss) equipped with a 514 nm laser line and a 20× 0.5 dry objective (EC Plan-Neofluar; Carl Zeiss). tdTomato fluorescent signals were detected at 552–651 nm.

Transgenic rice plant tissues were visualized using confocal laser-scanning microscopy (TCS SP8; Leica Microsystems) equipped with 405 nm and a pulsed white-light laser (WLL) lines and 20× multi-immersion objective lens (PL APO CS2 20×/0.75 IMM CORR HC; Leica Microsystems), a 40× water-immersion objective lens (PL APO CS2 40×/1.10 W CORR HCX; Leica Microsystems), and a 63× glycerol immersion objective lens (HC PL APO 63×/1.30 GLYC CORR CS2; Leica Microsystems). For mOrange fluorescence, images were captured at 550–600 nm after excitation at 543 nm with WLL. For SR2200 fluorescence, images were captured at 410–480 nm after excitation at 405 nm with a solid-state laser. After image acquisition, the images were processed using LASX software (Leica Microsystems).

The fluorescence images of immature gemmae and mouse brains were obtained with an upright two-photon excitation microscope (FVMPE-RS; Olympus) equipped with dual femtosecond pulse lasers (InSight DeepSee Dual-OL; Spectra-Physics) and a 25× 1.00 N.A. objective (XLSLPLN25XGMP; Olympus) and a 10× 0.6 N.A. objective (XLPLN10XSVMP; Olympus). The excitation wavelength for 4′,6-diamidino-2-phenylindole, GFP, and tdTomato was 700, 860, and 1040 nm, respectively. Image processing was performed using Imaris (Carl Zeiss) and ImageJ 1.53k (NIH).

**Statistics and reproducibility**. The statistical significance between two groups was evaluated using a two-sided Welch's *t*-test, whereas comparisons of multiple groups were assessed using the Tukey–Kramer method. The actual *p* values were superimposed on each graph. Boxplots show the median (middle bar), 25th and 75th percentiles (upper and lower limits of the box), and 1.5× interquartile range (whiskers). Each data point is represented by an open circle, and each mean is represented by a red cross. The sample sizes are indicated in the figure legends and Supplementary Data 1. Microsoft Excel (Microsoft) was used for all statistical analysis, and all graphs were plotted using the R studio software (https://www.rstudio.com).

**Reporting summary**. Further information on research design is available in the Nature Research Reporting Summary linked to this article.

## Data availability

All source data generated and/or analyzed during this study are included in this article (and its Supplementary Information files as Supplementary Data 1) or are available from the corresponding author on reasonable request. Plasmids have been deposited previously as pMpGWB116 (Acc. No. LC057458) and pMpGWB303 (Acc. No. LC057519)

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

## Acknowledgements

We thank Dr. Kimitsune Ishizaki for providing the histone H2B-tdTomato entry vector and Dr. Yuichiro Watanabe, Dr. Takahiro Hamada, Mr. Kazutaka Futagami, and Ms. Sakiko Ishida for technical support with transformation of *M. polymorpha*. This research was supported by MXT/JSPS KAKENHI 18K14743 and 21K06247 to Y. Sakamoto, 20H04884 to R.N., and 19H03259, 20H03297, and 20H05911 to S.M. Y. Sakamoto is also supported by the Osaka University Program for the Support of Networking among Present and Future Researchers, and S.M. is also supported by the Mitsubishi Foundation. We thank Edanz (https://jp.edanz.com/ac) for editing a draft of this manuscript.

## Author contributions

Y. Sakamoto and S.M. designed the research; Y. Sakamoto, A.I., Y. Sakai, M.S., R.N., K.A., and Y. Sano performed the research; Y. Sakamoto and A.I. analyzed the data; Y. Sakamoto, Y. Sano, T.F., H.T., T.K., and S.M. wrote the paper.

## Competing interests

The authors declare the following competing interests: the patent application in Japan. Patent applicant: Tokyo University of Science, Inventors: Yuki Sakamoto and Sachihiro Matsunaga, Application No.: 2020-026975, Status: pending. Yuki Sakamoto and Sachihiro Matsunaga are unpaid advisory positions of plant tissue-clearing reagents in Tokyo Chemical Industry Cooperation (Tokyo, Japan). The remaining authors declare no competing interests.
