## [Transparent Peer Review File · Communications Biology]

Reviewers' comments:

Reviewer #1 (Remarks to the Author):

In this manuscript, the authors developed iTOMEI, the improved version of their previous clearing method, TOMEI, for plant tissue clearing. The reason they developed iTOMEI was that TOMEI failed to remove chlorophylls entirely and to preserve the fluorescence signal of fluorescent proteins.

Overall, I found this manuscript quite difficult to read, mainly because, in many cases, although grammatically correct, the meaning of a sentence is often very ambiguous. For example, in line 46-47 they said "iTOMEI enables detection of much brighter FP fluorescence than previous methods within 26 h". After reading the manuscript, I know what they were trying to say is since iTOMEI enables better preservation of FP fluorescence than the previous one, thus one can observe much brighter fluorescence, but the sentence itself could mislead the reader. There are many more examples of this kind in the entire manuscript, which should be clarified.

The improvement of clearing performance by this method over the previous one is also limited. Moreover, the manuscript was carelessly prepared and necessary information was often omitted especially in figure legends.

In addition, no mention in the Method of how they perform the statistics. Indeed, the statistics they used are not appropriate. For example, in Fig 1b, e-j, they used unpaired two-sides t-test, but it should have been either ANOVA or a corresponding non-parametric test depending on the distribution of the samples. They should provide details about statistics in the Methods.

Details are below:

1. The most serious problem with this manuscript is that the author continued to change the conditions of iTOMEI during experiments. They said they optimized iTOMEI condition at 1% FA, 20% caprylyl sulfobetaine, 70.4% Iohexol (line 181-183). However, instead of using 1% FA, 2% FA was used in Fig.1d, and 4% FA was used in Fig. 4. In addition, 10% but not 20 % caprylyl sulfobetaine was used in Fig. 2c. Then, what is the purpose of the optimization? The author should explain why and should stick with the optimized condition throughout the manuscript.
2. In Fig 1c, they claimed that 1% FA preserved the best GFP fluorescence. But it seems that 2% and 4% FA show a much higher signal-to-background ratio, thus clearer images of nuclear speckles compared to 1% FA (See enlarged pictures). Rather than just saying sufficient or best, the authors should measure the signal-to-background ratio and compare it with each fixation condition. In line with this, why low mag pictures of 2% and 4% appear so dark as opposed to enlarged ones?
3. Line 128, they said " Both detergents slightly enhanced GFP fluorescence after incubation (Fig. 1e)." but Fig. 1e showed "significant (almost twice)" enhancement rather than "slightly". Besides, the y-axis is the fluorescence intensity in A.U., then, why is the fluorescence intensity so low between 5-20 A.U.?
4. Line 133, "To enhance the effect of a zwitterionic detergent,". The expression "To enhance the effect" is ambiguous. Enhance which effect of detergent?
5. Line 139-142, they said 20% caprylyl sulfobetaine showed the highest efficiency of chlorophyll elution and maintenance of GFP fluorescence but in Fig. 1i and j, no difference between 10 and 20 % caprylyl sulfobetaine in both measures.
6. In Fig. 1 legend, no explanations regarding f-i.
7. Line 153-156 & Fig.2b. which of three buffers was used for Fig.2b data?
8. Line 171. Fixed and decolorized seedlings... fixed how? decolorized how?
9. Fig. 3c. like Fig 1c, it seems iTOMEI shows a higher background signal. The authors should measure the SNB ratio and compare it with each method.
10. Image quality of Fig.4b and Fig.5b are poor (for example, Fig. 5b TOMEI-II image at 50 um is overly saturated).
11. Clearsee clearing was performed only for 26 hours. The author should perform FP comparison test

after complete clearing, for example, 4days as published previously (Development (2015) 142, 4168-4179 doi:10.1242/dev.127613))

12. In Figure 4c, besides PBS, Comparison with TOMEI-II is also required.

13. The superior of iTOMEI to TOMEI-II in clearing immature gemmae in Fig. 5c-f was not sufficiently demonstrated. Additional data comparing iTOMEI's performance with that of TOMEI-II are required to support their claim.

14. Mouse brain clearing experiment has nothing to do with the purpose of this study, thus should be discarded.

Minor concern;

1. Line 98-99, "...using GFP fluorescence, which is unstable under an acidic pH". The author should avoid using the word "unstable". Should be more specific such as decrease/quench etc.

2. Fig. 1i and j are not mentioned in the manuscript.

3. Poor organization of the complicated listing of various tissues and clearing methods could confuse the reader. The author should re-organize the Method section.

4. Line 249, Fig.5e should be Fig.5f.

Reviewer #2 (Remarks to the Author):

The manuscript titled "Improved clearing method contributes to deep imaging of plant organs" by Sakamoto et al. introduces a new clearing method named iTOMEI, an implementation of the TOMEI clearing method, to perform plant organs clearing. iTOMEI removes chlorophylls from plants using caprylyl sulfobetaine solution and restores endogenous fluorescence using a weak alkaline solution. The authors applied the clearing to different plants: *Arabidopsis thaliana*, *Oryza sativa*, and *Marchantia polymorpha*. Finally, they also demonstrate the clearing capability of iTOMEI on a mouse brain.

Following my observations:

Introduction:

1. 2,2'-thiodiethanol (TDE) is miscible with water and adjusted to a pH=7.6 does not produce endogenous fluorescence proteins quenching in mouse brains at concentrations below 80% TDE/PBS (as demonstrated by Aoyagi, Y. et al 2015 and Costantini et al 2015). Please consider adding this information in the introduction (line 85).

Results

1. Chapter 1-3: The endogenous fluorescence loss of FPs is attributed to formaldehyde fixation; however, the pH of the solution is not indicated by the authors. It is well known that low pH decreases the FP fluorescence rapidly up to completely quench it due to the protonation of the tyrosine present in the GFP chromophore. To restore the fluorescence it is necessary to increase the pH, as also the authors observed in the third chapter. pH equal to 7 is a value near to the starting of the quenching effect and could be the cause of it, instead of the fixation. To conclude that is the fixation protocol that induces FP quenching, authors should try to increment the pH of the fixative solutions to see if the fixative or the pH is the quencher of the fluorescence. Indeed, it was observed in other brain clearing studies that pH>7.5 maintains better the endogenous fluorescence of biological tissue, while the fluorescence increases with the pH (i.e Li et al. 2018 Front. Neuroanat.). The first and the third chapters of the results section need to be rewritten accordingly to these observations and the test results performed using solutions at different pH.

2. Chapter 4: Authors compare the refractive index matching effect of TDE 97% and iohexol 70.4% and they concluded that iohexol is better. However, based on the figure presented (fig 2e) the gray value is higher in TDE cleared leaf, not in those in iohexol. Please specify it in the text.

3. Add the reference to Figure 1j at line 139.

4. Please add the RI value matched with TDE 97% and iohexol 70.4%, I supposed it is 1.51-1.52

looking at the references mentioned by the authors, but it could be useful for the readers to have it in the paper.

5. Chapter 5: At lines 195-6-7 authors said: "Although GFP signals were detected at 75 μm depth in cotyledons cleared using ClearSee or iTOMEI, the brightest fluorescence at 75 μm depth in the cotyledon was detected after iTOMEI treatment." without providing any quantification of this observation. Please add it, since it is not obvious by-eye.

6. Chapter 6: In chapter 5 authors present a comparison with the ClearSee method, but they didn't in for the other two plans, why? Please give an explanation.

7. Line 205, please indicate the sectioning used for SAM analysis. Since the thickness of the sections is below 150 μm (from the information of line 204) it is not clear why it is necessary to apply a clearing method to perform the imaging. Could it not be performed directly on the cut sections? Please discuss it and add a fluorescence image in the supplementary materials demonstrating the unfeasibility of performing fluorescence imaging on not-cleared samples.

8. Line 247 as well as for the previous comment on confocal imaging please explain why it is necessary to combine two-photon fluorescence imaging (known to go deep inside the tissue) with clearing.

9. Chapter 7: tissue clearing is born on the mouse brain, I suggest putting Supplementary figure 2 in the main text to give it higher importance.

10. Line 268 "iTOMEI-B allowed acquisition of fluorescence images to 3 mm depth in transparent brain expressing GFP" is misleading, the brain sections are cut at 2mm, how it is possible to image up to 3mm? Please rewrite the sentence.

Discussion:

1. In Line 282 add the thickness of the samples analyzed in the study. Compared to brain clearing 150-200 μm of thickness is not consider "thick specimens" please remove it.

2. Line 290 FP quenching effect is attributed to fixation please consider modifying it depending on the results suggested above.

3. Line 320-323 the sentence "the transparency in the brain treated with SeeDB2 was lower compared with iTOMEI-B but the clearing brain by SeeDB2 will highly preserve the cell morphology and the fluorescence of FPs and minimizes the spherical aberration in super-resolution 3D imaging⁷" is misleading, it seems that iTOMEI could introduce morphological deformation while SeeDB2 not. If it is the case, here is the first time that this unwanted effect of iTOMEI is presented. There is no evaluation of morphological deformation introduced by iTOMEI in the text, please add it and discuss it properly.

4. Line 328 and 329: "thick organs" and "profound depth" are misleading, please add the thickness observed in the study: 200 μm .

Materials and Methods:

1. Please make explicit all the acronyms used in the study the first time you use them in the section.

2. Explain how the FA solution is prepared, it is bought, or made?

3. Line 451 explicit the "various detergent solutions" used.

4. Explain how the PBS solution is prepared and the concentration used.

5. Specify the proportion of the sodium phosphate buffer (line 461-462)

6. Line 477, please insert the information about the Tissue-Clearing Reagent TOMEI, the reader should know it without the need to go to the previous paper.

7. Line 485, specify the ClearSee concentration used.

8. Concerning the objectives used for the imaging, there are different RI used (air, water, glycerol) but there is no discussion of the spherical aberration introduced using them. Please add it.

Reviewer #3 (Remarks to the Author):

In this manuscript, authors developed a tissue-clearing method called "iTOMEI" for improved TOMEI for plant deep tissue imaging. The same group previously developed TOMEI, which uses high concentration of TDE. However, specimen cleared with TOMEI still contained chlorophyll, and the

fluorescent intensity of fluorescent proteins in the cleared specimen was reduced. In this manuscript, authors re-examined each step of TOMEI, and succeeded to establish a new protocol that produces specimen with less chlorophyll and enhanced fluorescence intensity of fluorescent proteins compared to TOMEI or other clearing method. Authors confirmed that iTOMEI can be applied for various plant materials including *Arabidopsis thaliana* seedlings, rice leaves, meristem and roots, and liverwort gemmaling. Using iTOMEI, authors found that MpRSL1 expresses in apical region of liverwort. Based on iTOMEI, authors developed iTOMEI-B for clearing mouse brain.

Overall, this manuscript contains clear data based on well-controlled experiments. As deep-tissue imaging is increasingly important method, this manuscript will be of great interest for many readers. However, there are many points that authors must revise before publication. Particularly, as this is a manuscript reporting a new methodology, authors should make sure that all the protocols were explicitly described. Specific comments are listed below.

Specific comments

(1) In Introduction, please describe more detailed protocol, merits and demerits of previously reported clearing methods for plants, ClearSee, PEA-CLATIRY and TOMEI. For example, in p5 L82 authors described "Scale-based methods, ClearSee and PEA-CLATIRY are slow to complete clearing", but it is not clear how long it actually takes to complete the process. Authors should include detailed information regarding the used reagents and time-frame to complete clearing for comparison of those methods. Please also describe what kind of plant samples were used for application of those methods and what was revealed by those methods.

(2) In Results, authors should first describe a detailed protocol for original TOMEI (preferably in the same format used to describe iTOMEI protocol in p9 L180-183) before describing the modification process in Results section to give readers an overall picture.

(3) In p5 L98-99, please give a reference for the description "GFP fluorescence, which is unstable under an acidic pH."

(4) In p5 L100, please describe the time taken for tissue fixation.

(5) In p6 L111~, it is not clear why weak fixation caused loss of nuclear speckles labeled with PCNA-GFP, while non-fixed live cells as well as sufficiently fixed cells showed nuclear speckles. Please explain.

(6) In p6 L121, which concentration of FA was used for fixing 2-week-old seedlings for chlorophyll removal experiments?

(7) In p7 L133, please explain what is zwitterionic detergent and what the effect is expected to be caused by zwitterionic detergent.

(8) In p7 L145 "Chemical reactivation enables recovery of reduced GFP fluorescence in resin-embedded specimens by alkaline treatment in animal tissues". Please refer appropriate papers.

(9) In p7 147, please describe how tissues were fixed.

(10) In p9 L187, authors must explain what TOMEI-II is, and the difference between TOMEI and TOMEI-II.

(11) In p9 L186~, authors compared samples prepared with TOMEI-II, ClearSee and iTOMEI. For this comparison, the total processing time was unified with 26 h. This description seems contradictory to that in Introduction, in which authors discussed that the downside of ClearSee is that this process takes time (assumingly compared to TOMEI). Overall, it is not clear the merits and demerits of iTOMEI compared to other clearing methods from this manuscript.

(12) Related to the comment 8, authors must discuss the merits or demerits of iTOMEI compared to other clearing methods such as ClearSee and PEA-CLARITY in Discussion section.

(13) In Materials and Methods, please give more detailed information for fluorescence intensity measurement. Did authors measure total fluorescence from the entire image, which contains fluorescent tissues and non-fluorescent background (such as shown in Fig. 1c PCNA-GFP)? Or, authors select a certain area in the image so that ROI contains only fluorescent tissues or cells? It is also not clear how relative fluorescence intensity was obtained. Did authors use the same area of same sample to compare before and after fixation, or use the average fluorescence intensity of several samples?

Point-by-point response to reviewers' comments

Reviewer #1 (Remarks to the Author):

In this manuscript, the authors developed iTOMEI, the improved version of their previous clearing method, TOMEI, for plant tissue clearing. The reason they developed iTOMEI was that TOMEI failed to remove chlorophylls entirely and to preserve the fluorescence signal of fluorescent proteins. Overall, I found this manuscript quite difficult to read, mainly because, in many cases, although grammatically correct, the meaning of a sentence is often very ambiguous. For example, in line 46-47 they said that iTOMEI enables detection of much brighter FP fluorescence than previous methods within 26 h. After reading the manuscript, I know what they were trying to say is since iTOMEI enables better preservation of FP fluorescence than the previous one, thus one can observe much brighter fluorescence, but the sentence itself could mislead the reader. There are many more examples of this kind in the entire manuscript, which should be clarified.

Response: Thank you for your valuable comment. We carefully checked and rewrote the manuscript.

The improvement of clearing performance by this method over the previous one is also limited. Moreover, the manuscript was carelessly prepared and necessary information was often omitted especially in figure legends.

Response: We quantified and compared the fluorescence intensity between iTOMEI and previous methods in Fig. 4b, d, e and Fig. 6d.

In addition, no mention in the Method of how they perform the statistics. Indeed, the statistics they used are not appropriate. For example, in Fig 1b, e-j, they used unpaired two-sides t-test, but it should have been either ANOVA or a corresponding non-parametric test depending on the distribution of the samples. They should provide details about statistics in the Methods.

Response: In accordance with your comment, we added "Statistics and reproducibility" in the Methods section as follows. "The statistical significance between two groups was evaluated using a two-sided Welch's *t*-test, whereas comparisons of multiple groups were assessed using the Tukey-Kramer method."

Details are below:

1. The most serious problem with this manuscript is that the author continued to change the conditions of iTOMEI during experiments. They said they optimized iTOMEI condition at 1% FA, 20% caprylyl sulfobetaine, 70.4% Iohexol (line 181-183). However, instead of using 1% FA, 2% FA was used in Fig.1d, and 4% FA was used in Fig. 4. In addition, 10% but not 20 % caprylyl sulfobetaine was used

in Fig. 2c. Then, what is the purpose of the optimization? The author should explain why and should stick with the optimized condition throughout the manuscript.

Response: We apologize for the ambiguous sentences. We modified the order of figures to avoid being misleading. In accordance with your comment, 20% caprylyl sulfobetaine was used in Fig. 2c.

2. In Fig 1c, they claimed that 1% FA preserved the best GFP fluorescence. But it seems that 2% and 4% FA show a much higher signal-to-background ratio, thus clearer images of nuclear speckles compared to 1% FA (See enlarged pictures). Rather than just saying sufficient or best, the authors should measure the signal-to-background ratio and compare it with each fixation condition. In line with this, why low mag pictures of 2% and 4% appear so dark as opposed to enlarged ones?

Response: PCNA forms nuclear speckles in the late S-phase; however, it localizes to the nucleoplasm throughout the cell cycle. Fluorescence in the nucleoplasm in the late S-phase is not background noise. To investigate signal-to-noise ratio (SNR) of all fixation conditions, signals were detected from the nucleoplasm, and the noise was detected from the cytoplasm. In Supplementary Fig. 1, 1% FA showed a high SNR compared with 0.1%, 2%, and 4% FA. In addition, low and high magnification images have the same brightness and contrast, as shown in Fig. 3c.

3. Line 128, they said “Both detergents slightly enhanced GFP fluorescence after incubation (Fig. 1e); but Fig. 1e showed significant (almost twice) enhancement rather than slightly”. Besides, the y-axis is the fluorescence intensity in A.U., then, why is the fluorescence intensity so low between 5-20 A.U.?

Response: Because the fluorescence images are 8 bit, the maximum value of the A.U. is 255. To avoid the saturation of signals, the photos were taken using microscope settings that do not allow the maximum value to exceed 100 A.U. in the photo. Consequently, the A.U. value is much lower than 255.

4. Line 133, “To enhance the effect of a zwitterionic detergent”. The expression “To enhance the effect” is ambiguous. Enhance which effect of detergent?

Response: In accordance with your comment, we modified the sentence as follows. “To enhance the effect of caprylyl sulfobetaine on chlorophyll clearance,”

5. Line 139-142, they said 20% caprylyl sulfobetaine showed the highest efficiency of chlorophyll elution and maintenance of GFP fluorescence but in Fig. 1i and j, no difference between 10 and 20 % caprylyl sulfobetaine in both measures.

Response: Thank you for your critical comment. We doubted that 10% caprylyl sulfobetaine was enough to elute chlorophyll from 14-day-old plants, because there were no differences when using

10%, 20%, and 30%, as shown in Fig. 1i, in the previous manuscript version. Consequently, we performed a similar experiment using 20-day-old plants (larger than 14-day-old ones), and the results showed that 20% caprylyl sulfobetaine eluted more chlorophyll than 10% caprylyl sulfobetaine, as shown in Fig. 1f.

6. In Fig. 1 legend, no explanations regarding f-i.

Response: We have added the appropriate explanations to the Figure 1 legend.

7. Line 153-156 & Fig.2b. which of three buffers was used for Fig.2b data?

Response: In accordance with your comment, we added an explanation.

8. Line 171. Fixed and decolorized seedlings; fixed how? decolorized how?

Response: We modified the sentence as “To confirm the impact of iohexol on GFP, plants expressing GFP were fixed with 2% FA in PBS buffer for 1 h and treated with 20% (w/v) caprylyl sulfobetaine solution for 24 h.”

9. Fig. 3c. like Fig 1c, it seems iTOMEI shows a higher background signal. The authors should measure the SNB ratio and compare it with each method.

Response: In accordance with your comment, we measured the signal-to-noise ratio in Fig. 4e.

10. Image quality of Fig.4b and Fig.5b are poor (for example, Fig. 5b TOMEI-II image at 50 um is overly saturated).

Response: The brightness and contrast levels in Fig. 5b and Fig. 6b were changed to avoid signal saturation. However, the Fig. 6b image at 50 μ m is still saturated because the laser and detector settings were the same for all the images. The laser power to observe deep tissues (for example, the image at 200 μ m) is too strong to observe shallow areas, and the signals are saturated in shallow areas.

11. Clearsee clearing was performed only for 26 hours. The author should perform FP comparison test after complete clearing, for example, 4days as published previously (Development (2015) 142, 4168-4179 doi:10.1242/dev.127613))

Response: Thank you for your critical comment. In accordance with your comment, we performed ClearSee for 4 days and modified Figure 3.

12. In Figure 4c, besides PBS, Comparison with TOMEI-II is also required.

Response: In accordance with your comment, we added a TOMEI-II image to Fig. 5c.

13. The superior of iTOMEI to TOMEI-II in clearing immature gemmae in Fig. 5c-f was not sufficiently demonstrated. Additional data comparing iTOMEI's performance with that of TOMEI-II are required to support their claim.

Response: We added the quantification data to Fig. 6d to show that iTOMEI is more effective than TOMEI. The fluorescence intensity at a depth of 100–110 μm was quantified for each of the three treatments.

14. Mouse brain clearing experiment has nothing to do with the purpose of this study, thus should be discarded.

Response: We want to keep the mouse brain data in the supplementary materials because Reviewer #2 said that these data are essential.

Minor concern;

1. Line 98-99, "using GFP fluorescence, which is unstable under an acidic pH"; The author should avoid using the word "unstable". Should be more specific such as decrease/quench etc.

Response: In accordance with your comment, we modified the sentence.

2. Fig. 1i and j are not mentioned in the manuscript.

Response: In accordance with your comment, we have referenced Fig. 1f and 1g in the manuscript.

3. Poor organization of the complicated listing of various tissues and clearing methods could confuse the reader. The author should re-organize the Method section.

Response: In accordance with your comment, we modified the Methods section.

4. Line 249, Fig.5e should be Fig.5f.

Response: We have changed Fig. 5e to Fig. 5f.

Reviewer #2 (Remarks to the Author):

The manuscript titled "Improved clearing method contributes to deep imaging of plant organs" by Sakamoto et al. introduces a new clearing method named iTOMEI, an implementation of the TOMEI clearing method, to perform plant organs clearing. iTOMEI removes chlorophylls from plants using caprylyl sulfobetaine solution and restores endogenous fluorescence

using a weak alkaline solution. The authors applied the clearing to different plants: *Arabidopsis thaliana*, *Oryza sativa*, and *Marchantia polymorpha*. Finally, they also demonstrate the clearing capability of iTOMEI on a mouse brain.

Following my observations:

Introduction:

1. 2,2-thiodiethanol (TDE) is miscible with water and adjusted to a pH=7.6 does not produce endogenous fluorescence proteins quenching in mouse brains at concentrations below 80% TDE/PBS (as demonstrated by Aoyagi, Y. et al 2015 and Costantini et al 2015). Please consider adding this information in the introduction (line 85).

Response: In accordance with your comment, we added the information on TDE in the introduction section.

Results

1. Chapter 1-3: The endogenous fluorescence loss of FPs is attributed to formaldehyde fixation; however, the pH of the solution is not indicated by the authors. It is well known that low pH decreases the FP fluorescence rapidly up to completely quench it due to the protonation of the tyrosine present in the GFP chromophore. To restore the fluorescence it is necessary to increase the pH, as also the authors observed in the third chapter. pH equal to 7 is a value near to the starting of the quenching effect and could be the cause of it, instead of the fixation. To conclude that is the fixation protocol that induces FP quenching, authors should try to increment the pH of the fixative solutions to see if the fixative or the pH is the quencher of the fluorescence. Indeed, it was observed in other brain clearing studies that pH>7.5 maintains better the endogenous fluorescence of biological tissue, while the fluorescence increases with the pH (i.e Li et al. 2018 *Front. Neuroanat.*).

The first and the third chapters of the results section need to be rewritten accordingly to these observations and the test results performed using solutions at different pH.

Response: We measured GFP fluorescence after fixation using phosphate buffers at pH 7.0, 7.5, and 8.0, but we did not detect any difference among the three buffers, as shown in Supplemental Fig. 2. We added the pH levels of the fixation buffers to the Methods section.

2. Chapter 4: Authors compare the refractive index matching effect of TDE 97% and iohexol 70.4% and they concluded that iohexol is better. However, based on the figure presented (fig 2e) the gray value is higher in TDE cleared leaf, not in those in iohexol. Please specify it in the text.

Response: In accordance with your comment, we added the sentence as “Although TDE made the leaves slightly more transparent than iohexol, both TDE and iohexol decreased the intensity of the

grids behind the leaves compared with PBS buffer (Fig. 2e).”

3. Add the reference to Figure 1j at line 139.

Response: We added the reference.

4. Please add the RI value matched with TDE 97% and iohexol 70.4%, I supposed it is 1.51-1.52 looking at the references mentioned by the authors, but it could be useful for the readers to have it in the paper.

Response: In accordance with your comment, we added the RI values.

5. Chapter 5: At lines 195-6-7 authors said: “Although GFP signals were detected at 75 μ m depth in cotyledons cleared using ClearSee or iTOMEI, the brightest fluorescence at 75 μ m depth in the cotyledon was detected after iTOMEI treatment; without providing any quantification of this observation. Please add it, since it is not obvious by-eye.

Response: In accordance with your comment, we added the quantification data to Fig. 4d.

6. Chapter 6: In chapter 5 authors present a comparison with the ClearSee method, but they didn't in for the other two plans, why? Please give an explanation.

Response: We believe that the comparison between iTOMEI and ClearSee was enough in Fig. 4. Because we wanted to examine the adaptability of iTOMEI to rice and *M. polymorpha* in Figs. 5 and 6, respectively, it was not necessary to perform ClearSee in these plants.

7. Line 205, please indicate the sectioning used for SAM analysis. Since the thickness of the sections is below 150 μ m (from the information of line 204) it is not clear why it is necessary to apply a clearing method to perform the imaging. Could it not be performed directly on the cut sections? Please discuss it and add a fluorescence image in the supplementary materials demonstrating the unfeasibility of performing fluorescence imaging on not-cleared samples.

Response: We apologize for the ambiguous sentence. We observed SAM without sectioning in this experiment. To avoid being misleading, we added the following sentence, “ We attempted to observe FPs in the SAM using iTOMEI without sectioning.” and a schematic diagram in Supplementary Fig. 3.

8. Line 247 as well as for the previous comment on confocal imaging please explain why it is necessary to combine two-photon fluorescence imaging (known to go deep inside the tissue) with clearing.

Response: We had to determine whether iTOMEI could work in a system with a two-photon excitation microscope. If we determined that these two methods could be combined, then methods could be

developed to observe the deepest regions of plant organs.

9. Chapter 7: tissue clearing is born on the mouse brain, I suggest putting Supplementary figure 2 in the main text to give it higher importance.

Response: Because Reviewer #1 believed that these data are unimportant, they have remained in Supplementary Fig. 2.

10. Line 268 iTOMEI-B allowed acquisition of fluorescence images to 3 mm depth in transparent brain expressing GFP; is misleading, the brain sections are cut at 2mm, how it is possible to image up to 3mm? Please rewrite the sentence.

Response: We apologize for the ambiguous sentence. In this experiment (Supplementary Fig. 5b-d), we used the hemi-brains of mice. We rewrote the sentence as “Hemi brains expressing EGFP were also treated with PBS, SeeDB2, and iTOMEI-B, and observed from the cortical surface of the brain using a two-photon excitation microscope.”

Discussion:

1. In Line 282 add the thickness of the samples analyzed in the study. Compared to brain clearing 150-200 um of thickness is not consider thick specimens; please remove it.

Response: In accordance with your comment, we modified the sentence as follows: “This enables the three-dimensional detection of FP signals from a 200- μ m depth in plant tissues.”

2. Line 290 FP quenching effect is attributed to fixation please consider modifying it depending on the results suggested above.

Response: Because the pH of the fixation solution did not affect GFP fluorescence, we did not modify the sentence.

3. Line 320-323 the sentence the transparency in the brain treated with SeeDB2 was lower compared with iTOMEI-B but the clearing brain by SeeDB2 will highly preserve the cell morphology and the fluorescence of FPs and minimizes the spherical aberration in super-resolution 3D imaging; is misleading, it seems that iTOMEI could introduce morphological deformation while SeeDB2 not. If it is the case, here is the first time that this unwanted effect of iTOMEI is presented. There is no evaluation of morphological deformation introduced by iTOMEI in the text, please add it and discuss it properly.

Response: We apologize for the ambiguous sentence. We did not detect a morphological deformation in the mouse brain after the iTOMEI treatment. We modified the sentence as follows: “ iTOMEI-B achieved high transparency and bright GFP detection in the brain compared with SeeDB2 and revealed

minor alterations in the brain volume comparable with those observed using SeeDB2 (Supplementary Fig. 5a).”

4. Line 328 and 329: “thick organs” and “profound depth” are misleading, please add the thickness observed in the study: 200um.

Response: We modified the sentence as “Our developed iTOMEI is a powerful technique to transparentize plant organs with almost no attenuation of the fluorescence from FPs, which were expressed in the 200 μm depth of organs.”

Materials and Methods:

1. Please make explicit all the acronyms used in the study the first time you use them in the section.

Response: In accordance with your comment, we modified the manuscript.

2. Explain how the FA solution is prepared, it is bought, or made?

Response: We added the sentence as “One-week-old seedlings expressing GFP were fixed with 2% FA (Polyscience) in PBS buffer for 1 h. Since .” in the Methods section.

3. Line 451 explicit the “various detergent solutions” used.

Response: In accordance with your comment, we modified the sentence.

4. Explain how the PBS solution is prepared and the concentration used.

Response: In accordance with your comment, we added the sentence as “The PBS buffer (137 mmol/l NaCl, 8.1 mmol/l Na₂HPO₄, 2.68 mmol/l KCl, and 1.47 mmol/l KH₂PO₄, pH7.4) was prepared in the laboratory.” to the Methods section.

5. Specify the proportion of the sodium phosphate buffer (line 461-462)

Response: In accordance with your comment, we added the proportion of the sodium phosphate buffer.

6. Line 477, please insert the information about the Tissue-Clearing Reagent TOMEI, the reader should know it without the need to go to the previous paper.

Response: In accordance with your comment, we added the TOMEI contents.

7. Line 485, specify the ClearSee concentration used.

Response: In accordance with your comment, we added the ClearSee solution contents.

8. Concerning the objectives used for the imaging, there are different RI used (air, water, glycerol) but

there is no discussion of the spherical aberration introduced using them. Please add it.

Response: In accordance with your comment, we considered the RI values in the Discussion section.

Reviewer #3 (Remarks to the Author):

In this manuscript, authors developed a tissue-clearing method called 'iTOMEI' for improved TOMEI for plant deep tissue imaging. The same group previously developed TOMEI, which uses high concentration of TDE. However, specimen cleared with TOMEI still contained chlorophyll, and the fluorescent intensity of fluorescent proteins in the cleared specimen was reduced. In this manuscript, authors re-examined each step of TOMEI, and succeeded to establish a new protocol that produces specimen with less chlorophyll and enhanced fluorescence intensity of fluorescent proteins compared to TOMEI or other clearing method. Authors confirmed that iTOMEI can be applied for various plant materials including *Arabidopsis thaliana* seedlings, rice leaves, meristem and roots, and liverwort gemmaling. Using iTOMEI, authors found that *MpRSL1* expresses in apical region of liverwort. Based on iTOMEI, authors developed iTOMEI-B for clearing mouse brain. Overall, this manuscript contains clear data based on well-controlled experiments. As deep-tissue imaging is increasingly important method, this manuscript will be of great interest for many readers. However, there are many points that authors must revise before publication. Particularly, as this is a manuscript reporting a new methodology, authors should make sure that all the protocols were explicitly described. Specific comments are listed below.

Specific comments

(1) In Introduction, please describe more detailed protocol, merits and demerits of previously reported clearing methods for plants, ClearSee, PEA-CLATIRY and TOMEI. For example, in p5 L82 authors described 'Scale-based methods, ClearSee and PEA-CLATIRY are slow to complete clearing', but it is not clear how long it actually takes to complete the process. Authors should include detailed information regarding the used reagents and time-frame to complete clearing for comparison of those methods. Please also describe what kind of plant samples were used for application of those methods and what was revealed by those methods.

Response: In accordance with your comment, we added the information of clearing methods for plant tissues in the Introduction section.

(2) In Results, authors should first describe a detailed protocol for original TOMEI (preferably in the same format used to describe iTOMEI protocol in p9 L180-183) before describing the modification process in Results section to give readers an overall picture.

Response: In accordance with your comment, we added the TOMEI-II protocol in the first paragraph

of the Results section.

(3) In p5 L98-99, please give a reference for the description of GFP fluorescence, which is unstable under an acidic pH.

Response: We deleted this sentence.

(4) In p5 L100, please describe the time taken for tissue fixation.

Response: In accordance with your comment, we added the time.

(5) In p6 L111~, it is not clear why weak fixation caused loss of nuclear speckles labeled with PCNA-GFP, while non-fixed live cells as well as sufficiently fixed cells showed nuclear speckles. Please explain.

Response: In accordance with your comment, we added the sentence and references as follows: "Insufficient or inappropriate fixation may disrupt or alter the cellular structures^{22,23}"

(6) In p6 L121, which concentration of FA was used for fixing 2-week-old seedlings for chlorophyll removal experiments?

Response: In accordance with your comment, we added the concentration of FA.

(7) In p7 L133, please explain what is zwitterionic detergent and what the effect is expected to be caused by zwitterionic detergent.

Response: To avoid being misleading, we modified the sentence as follows "To enhance the effect of caprylyl sulfobetaine on chlorophyll clearance, we tested the combinations of caprylyl sulfobetaine with Triton X-100 used in the Scale-based method (Fig. 1c), with sodium deoxycholate used in the ClearSee (Fig. 1d), and with urea used in the Scale-based method and ClearSee (Fig. 1e)."

(8) In p7 L145 Chemical reactivation enables recovery of reduced GFP fluorescence in resin-embedded specimens by alkaline treatment in animal tissues. Please refer appropriate papers.

Response: In accordance with your comment, we added the references.

(9) In p7 147, please describe how tissues were fixed.

Response: In accordance with your comment, we added the fixation protocol.

(10) In p9 L187, authors must explain what TOMEI-II is, and the difference between TOMEI and TOMEI-II.

Response: We explained the difference between TOMEI-I and TOMEI-II in the Introduction section.

(11) In p9 L186~, authors compared samples prepared with TOMEI-II, ClearSee and iTOMEI. For this comparison, the total processing time was unified with 26 h. This description seems contradictory to that in Introduction, in which authors discussed that the downside of ClearSee is that this process takes time (assumingly compared to TOMEI). Overall, it is not clear the merits and demerits of iTOMEI compared to other clearing methods from this manuscript.

Response: Thank you for your critical comment. We performed ClearSee for 4 days to compare it with iTOMEI for 27 h, as shown in Fig. 4.

(12) Related to the comment 8, authors must discuss the merits or demerits of iTOMEI compared to other clearing methods such as ClearSee and PEA-CLARITY in Discussion section.

Response: In accordance with your comment, we added a comparison between iTOMEI and the other methods in the first paragraph of the Discussion section.

(13) In Materials and Methods, please give more detailed information for fluorescence intensity measurement. Did authors measure total fluorescence from the entire image, which contains fluorescent tissues and non-fluorescent background (such as shown in Fig. 1c PCNA-GFP)? Or, authors select a certain area in the image so that ROI contains only fluorescent tissues or cells? It is also not clear how relative fluorescence intensity was obtained. Did authors use the same area of same sample to compare before and after fixation, or use the average fluorescence intensity of several samples?

Response: In accordance with your comment, we added the information on measuring the fluorescence intensity in the Methods section.

Reviewers' comments:

Reviewer #1 (Remarks to the Author):

I read the revised manuscript and response letter carefully. I understand the authors should have added a substantial amount of time and effort to revise the manuscript to answer the reviewers' comments. Nevertheless, I have raised several issues, but some of the concerns still remain unanswered.

The revised manuscript is still quite hard to read and contains many ambiguous descriptions and inconsistencies. It should be carefully read and edited by the native speaker who is an expert in the biological science field.

Just name a few

Introduction "line 103. All of these methods..." this sentence should be moved to somewhere or be reorganized unless "these" mean TOMEI and II but not all other clearing methods.

Line 108:" the fluorescence intensity levels of the FPs".. should be "fluorescence intensity of the FPs". They have never defined the levels of fluorescence intensity.

And more can be found throughout the manuscript.

Regarding statistics, although the author included the Statistics section in the Methods, they described "The statistical significance between two groups was evaluated using a two-sided Welch's t-test, whereas comparisons of multiple groups were assessed using the Tukey-Kramer method."

But in fig legends (fig.1, 2, 4, 6), they said "...evaluated using the two-sided Welch's t-test." even for the comparisons of multiple groups.

To my previous question regarding the poor quality of images and saturation, they responded that the fluorescence images are 8 bit. I don't understand why they took pictures as 8-bit images. They said they used confocal microscopes (FV100, 1200 from Olympus, TCS SP8 from Leica, LSM710 from Zeiss etc) with 20, 40, or 63 x objective lens. All of them usually produce images of more than 8 bit (for example, Olympus FV1200, up to 2048 x 2048 pixel resolution and 12-bit grayscale can be obtained and LSM710 can produce 12 or 16-bit depth images). If pixels are saturated then, the feature could become very bright, appear to flatten, and may expand in size. It is the rule of thumb that the artifact should be avoided by utilizing a camera and software package that will allow greater bit depth and an algorithm that internally stores data as floating-point numbers.

Besides, 8-bit format is usually unsuitable for quantifying experimental image data since some of the fine detail in the image data can be lost.

Reviewer #2 (Remarks to the Author):

The manuscript has been greatly improved in this submission, the authors have satisfactorily addressed my comments and concerns. I recommend publishing the manuscript.

Reviewer #3 (Remarks to the Author):

Authors have fully revised the manuscript in response to reviewers' comments, which has much improved the manuscript. I have only two comments as described below.

1. Some of the newly added paragraphs (for example the first paragraph in the Introduction and the first paragraph of the Results "Caprylyl sulfobetaine elute chlorophylls without GFP quenching) are too long and should be divided into two or three paragraphs for readability.
2. In L100-101, the sentence "However, the final TDE concentration should depend on FP, because below 80% the TDE solution has little effect on FP fluorescence" does not make sense. It is not clear why the final TDE concentration should depend on FP. In addition, it is not clear what the "effect" given to FP fluorescence is.

Point-by-point responses to reviewers' comments

Reviewer #1:

I read the revised manuscript and response letter carefully. I understand the authors should have added a substantial amount of time and effort to revise the manuscript to answer the reviewers' comments.

Response: Thank you for high evaluation of our revisions.

Nevertheless, I have raised several issues, but some of the concerns still remain unanswered.

The revised manuscript is still quite hard to read and contains many ambiguous descriptions and inconsistencies. It should be carefully read and edited by the native speaker who is an expert in the biological science field.

Response: In accordance with your advice, we asked a native speaker with a PhD in this field to edit our manuscript.

1. Introduction "line 103. All of these methods..." this sentence should be moved to somewhere or be reorganized unless "these" mean TOMEI and II but not all other clearing methods.

Response: "All of these methods" means our described four different clearing methods. Thus, we revised the sentence as follows. "All four methods substantially ameliorated the optical properties of the specimens; however, each method has drawbacks."

2. Line 108: "the fluorescence intensity levels of the FPs" should be "fluorescence intensity of the FPs". They have never defined the levels of fluorescence intensity.

Response: In accordance with your advice, we deleted "levels" from the sentence.

3. Regarding statistics, although the author included the Statistics section in the Methods, they described "The statistical significance between two groups was evaluated using a two-sided Welch's t-test, whereas comparisons of multiple groups were assessed using the Tukey-Kramer method." But in fig legends (fig.1, 2, 4, 6), they said "...evaluated using the two-sided Welch's t-test." even for the comparisons of multiple groups.

Response: This comment is derived from a misunderstanding. Please look in the straight lines between two samples as shown in the graphs in Figs. 1, 2, 4, and 6. Because we are making a comparison between two samples connected by the straight line, the use of the two-sided Welch's t-test for the statistical analysis is perfectly acceptable. We did not perform any statistical analyses of multiple groups shown in the graphs of Figs. 1, 2, 4, and 6. We believe that you did not understand that the lines between two samples indicated samples that they were being compared, with their statistical

values being written above each line. To more clearly indicate the two samples used in the comparisons, we have added a downward demarcation at each end of each line to indicate the compared samples.

4. To my previous question regarding the poor quality of images and saturation, they responded that the fluorescence images are 8 bit. I don't understand why they took pictures as 8-bit images. They said they used confocal microscopes (FV100, 1200 from Olympus, TCS SP8 from Leica, LSM710 from Zeiss etc) with 20, 40, or 63 x objective lens. All of them usually produce images of more than 8 bit (for example, Olympus FV1200, up to 2048 x 2048 pixel resolution and 12-bit grayscale can be obtained and LSM710 can produce 12 or 16-bit depth images). If pixels are saturated then, the feature could become very bright, appear to flatten, and may expand in size. It is the rule of thumb that the artifact should be avoided by utilizing a camera and software package that will allow greater bit depth and an algorithm that internally stores data as floating-point numbers. Besides, 8-bit format is usually unsuitable for quantifying experimental image data since some of the fine detail in the image data can be lost.

Response: This comment is derived from a misunderstanding. Regarding your question about the fluorescence quantification of Figure 1e (Figure 1b in the present manuscript), we previously responded that 8-bit images were used for this analysis. Figure 1 shows the screening process used to identify suitable substances for the improvement of our transparency method. During this process, we analyzed the 8-bit images acquired by a fluorescence stereomicroscope (SMZ18; Nikon) equipped with a DS-Ri2 digital camera (Nikon) or a fluorescence stereomicroscope (SZX16; Olympus) equipped with a DP21 camera (Olympus), not confocal microscopes, as described in both the Results and Materials and Methods sections. The images were of high enough quality to evaluate or compare the effects of each substance in our screening steps. Except for these steps, we acquired 12-bit images using confocal microscopes, which are shown in the figures.

Reviewer #3:

Authors have fully revised the manuscript in response to reviewers' comments, which has much improved the manuscript.

Response: Thank you for high evaluation for our revisions.

1. Some of the newly added paragraphs (for example the first paragraph in the Introduction and the first paragraph of the Results "Caprylyl sulfobetaine elute chlorophylls without GFP quenching) are too long and should be divided into two or three paragraphs for readability.

Response: In accordance with your suggestions, we divided the first paragraphs in the Introduction

and Results sections into three and two paragraphs, respectively.

2. In L100-101, the sentence “However, the final TDE concentration should depend on FP, because below 80% the TDE solution has little effect on FP fluorescence” does not make sense. It is not clear why the final TDE concentration should depend on FP. In addition, it is not clear what the “effect” given to FP fluorescence is.

Response: In accordance with your advice, we revised the sentence as follows:

“The final TDE concentration should be considered carefully because a solution of less than 80% (v/v) TDE has a limited quenching effect on FPs¹⁶⁻¹⁸, whereas a solution of greater than 80% (v/v) TDE largely quenches FPs.”